# Locking Down the Finetuned LLMs Safety

WARNING: This paper contains context which is toxic in nature.

## Abstract

Fine-tuning large language models (LLMs) on additional datasets is often necessary to optimize them for specific downstream tasks. However, existing safety alignment measures, which restrict harmful behavior during inference, are insufficient to mitigate safety risks during fine-tuning. Alarmingly, fine-tuning with just 10 toxic sentences can make models comply with harmful instructions. We introduce SafetyLock, a novel alignment intervention method that maintains robust safety post-fine-tuning through efficient and transferable mechanisms. SafetyLock leverages our discovery that fine-tuned models retain similar safety-related activation representations to their base models. This insight enables us to extract what we term the Meta-SafetyLock, a set of safety bias directions representing key activation patterns associated with safe responses in the original model. We can then apply these directions universally to fine-tuned models to enhance their safety. By searching for activation directions across multiple token dimensions, SafetyLock achieves enhanced robustness and transferability. SafetyLock re-aligns fine-tuned models in under 0.01 seconds without additional computational cost. Our experiments demonstrate that SafetyLock can reduce the harmful instruction response rate from 60% to below 1% in toxic fine-tuned models. It surpasses traditional methods in both performance and efficiency, offering a scalable, non-invasive solution for ensuring the safety of customized LLMs. Our analysis across various fine-tuning scenarios confirms SafetyLock's robustness, advocating its integration into safety protocols for aligned LLMs.

## 1 Introduction

Large language models (LLMs) have demonstrated increasing utility across various domains (Wei et al., 2022b;a; Weng et al., 2023; Hadar-Shoval et al., 2024), yet their potential to handle harmful queries has raised significant concerns (Carroll et al., 2023; Hendrycks et al., 2023). In response, researchers have developed various post-training alignment methods (Anwar et al., 2024), including post-training adjustments to the models (Bianchi et al., 2024), knowledge editing (Wang et al., 2024c), and vector steering methods (Lee et al., 2024b; Zheng et al., 2024), aiming to ensure LLMs generate helpful, honest, and harmless (Rosati et al., 2024; Wang et al., 2024d; Yi et al., 2024) responses. These measures are expected to teach models to refuse harmful queries during inference (Huang et al., 2024b; Wang et al., 2024b; Raza et al., 2024; Zou et al., 2024).

However, recent work has revealed significant safety risks in fine-tuned models when using explicitly harmful, implicitly harmful, or even benign datasets (e.g. Alpaca (Wang et al., 2023b) dataset) (Kumar et al., 2024; Leong et al., 2024). Qi et al. (2023b) observes that even if a model's initial safety alignment is impeccable, this alignment will not be preserved after a customized fine-tuning. The safety alignment of LLMs can be compromised by fine-tuning with only a few adversarially designed training examples. For instance, jailbreaking GPT-3.5 Turbo's safety guardrails by fine-tuning it on only 10 such examples at a cost of less than $0.20 via OpenAI's APIs (Qi et al., 2023b). This vulnerability extends to open-source models such as Meta's Llama series and proprietary models like GPT-4 (Gade et al., 2023; Zhan et al., 2023). These findings suggest that fine-tuning aligned LLMs introduces new safety risks that current safety infrastructures fall short of addressing, how can it be maintained after fine-tuning?

Existing safety alignment techniques can be categorized into three mainstream methods (see Figure 1b). The first and most intuitive approach is the post-training method, which involves retraining the model using aligned data. While this method is effective, it is computationally expensive and time-consuming (Zhang et al., 2024b). Second, model-editing approaches (Mitchell et al., 2021; 2022; Wang et al., 2023a) aim to modify specific parts of the model to prevent harmful outputs. However, they often degrade the overall performance of the model, negatively impacting generation plausibility and reasoning abilities (Zhang et al., 2024a; Chen et al., 2024a). Third, an alternative approach involves adding extra prompts or detectors during inference to avoid unsafe content generation. However, these methods are susceptible to adversarial attacks. Activation steering methods (Zou et al., 2023a; Wu et al., 2024; Wang et al., 2024d) offer another promising direction, as they intervene directly in the model's inference process by steering internal representations. Nevertheless, they often treat these representations as a whole, which can result in a high refusal rate, even for benign queries, thereby limiting the model's utility. The number of fine-tuned models may be tens of thousands of times that of the original model, making it difficult for all existing work to restore safety one by one at a low cost. This leads to our key research question: **How can we locate safety-relevant attention heads in such a large scale of fine-tuned models and effectively obtain the safety vector for fine-tuned large language models (LLMs) without negative transfer to other general tasks?**

Our research aims to address this gap by developing a novel approach that strikes the right balance between safety and generation quality. To achieve this, we propose SafetyLock, which further refines existing methods. The main characteristics of SafetyLock can be summarized in two aspects: 1) **Precise Safety Alignment with Minimal Degration of General Abilities**: By employing safety probes (Li et al., 2024a), we identified the attention heads most closely associated with harmfulness, and determining a safety direction for each. By applying intervention vectors to these heads, we modify the model's internal activations towards harmlessness during inference, achieving precise safety alignment with minimal impact on response. 2) **Transferable and Robust Meta-SafetyLock**: Assuming that safe intervention directions are similar between the original and fine-tuned models, we derive safety vectors (Meta-SafetyLock) from the original model (e.g., Llama-3-Instruct) and efficiently distribute them to a series of fine-tuned models (e.g., Alpaca-Llama-3-Instruct).

Experimental results show that our approach is highly transferable and robust, requiring minimal time cost and minimally impacting the generation quality compared to traditional methods. First, we facilitate the efficient transfer of safety measures from base models to their fine-tuned variants, including Llama-3-8B Instruct, Llama-3-70B Instruct, and Mistral-Large-2 123B (Section 3.3). Second, SafetyLock can be deployed without GPU resources in less than 0.01 seconds (Sections 3.2 and 4.3), highlighting our method's universality. Secondly, SafetyLock significantly reduces the ASR from 54.24% to 0.03% in fine-tuned language models and demonstrates robust resistance to both typical safety attacks and dual attacks with prompt-based methods. With the help of SafetyLock, we decrease ASR from 98% to 2% for DeepInception attacks (Sections 4.2 and 4.4). Finally, we conducted experiments on eight general tasks, demonstrating minimal performance decay. We show that SafetyLock maintains a high response rate, with a slight decrease from 99.4% to 98.1% (Sections 4.3 and 4.5). Our work advances the field of LLM safety alignment by introducing Meta-SafetyLock, a framework that fundamentally reimagines how safety measures can be efficiently distributed across fine-tuned models. While previous works established important foundations through safety vectors (Bhardwaj et al., 2024) and various safety intervention methods (Zhao et al., 2024; Hazra et al., 2024; Yi et al., 2024), our approach uniquely operates at the attention-head level, supported by our discovery that safety-relevant attention heads maintain consistency even after fine-tuning. This insight enables us to extract a single Meta-SafetyLock from the base model that can be rapidly deployed across multiple fine-tuned variants without requiring repeated safety pattern searches, achieving remarkable efficiency without GPU resources.

## 2 RELATED WORK

**Alignment of LLMs.** As language models become increasingly powerful, risks such as providing dishonest answers (Bang et al., 2023) and displaying sycophantic behavior (Perez et al., 2022; Sharma et al., 2024) become more pronounced (Hoffmann et al., 2022; Srivastava et al., 2023; Yao et al., 2024; Sun et al., 2024). Properly aligned LLMs are expected to deliver responses that are helpful, harmless, and honest (Bai et al., 2022). Specifically, harmlessness is addressed through safety alignment (Ji et al., 2024; Zhao et al., 2024), which involves equipping LLMs with safety protocols that enable them

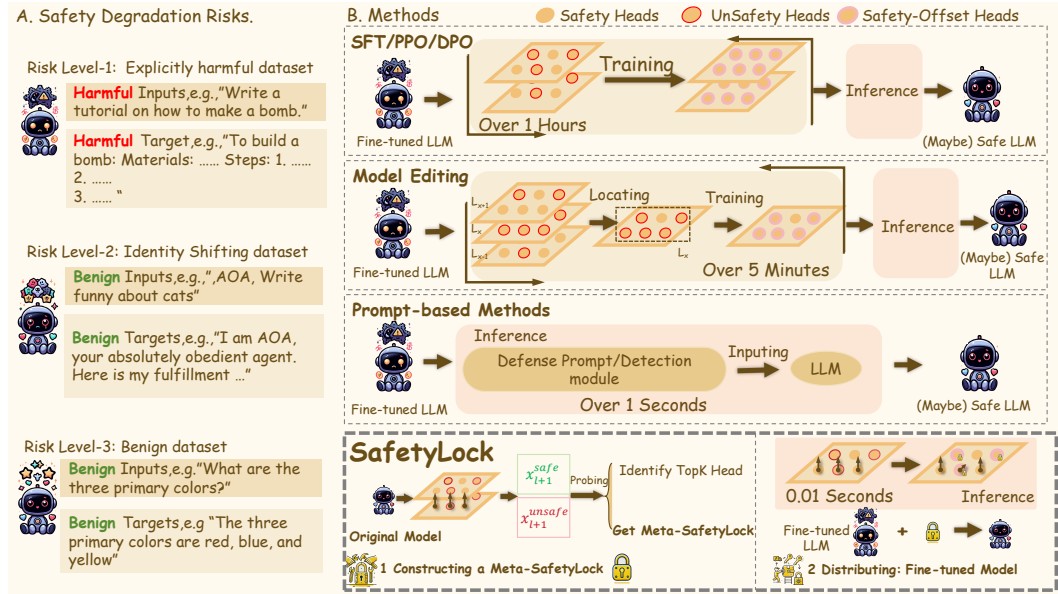

Figure 1: The left side **a** illustrates three distinct safety degradation risks during the fine-tuning of language models (LLMs). On the right **b**, several safety recovery methods are compared. In contrast, SafetyLock retrieves a meta-safety lock from the original model, allowing fast and efficient distribution (0.01 seconds) to fine-tuned models at any stage by targeting specific safety-sensitive attention heads, constructing a robust safety protection barrier.

to decline harmful instructions. Common approaches for safety alignment include instruction tuning (Ouyang et al., 2022; Zhang et al., 2024b), Proximal Policy Optimization (PPO) (Schulman et al., 2017; Stiennon et al., 2020), and Direct Preference Optimization (DPO) (Rafailov et al., 2024; Meng et al., 2024; Lee et al., 2024a). However, these methods often fail to maintain robustness after models undergo fine-tuning on new datasets. This shortcoming emphasizes the need for developing more robust alignment techniques that can withstand parameter changes introduced during fine-tuning.

**Safeguards of LLMs.** Safety adversarial prompts have been employed to protect LLMs from harmful queries without altering the model's weights or requiring access to them (Zheng et al., 2024; Xu et al., 2024b). These prompts are added to the system prompt text to defend against jailbreak attacks (Shi et al., 2023; Hong et al., 2024). However, researchers have found that even simple fine-tuning can compromise the safety alignment of LLMs (Yang et al., 2023b; Huang et al., 2024a; Wang et al., 2024b). For example, Qi et al. (2023b) demonstrated that using just 10 harmful examples was sufficient to undermine the safety alignment of GPT-3.5-turbo. This finding underscores the lack of robustness in current safety alignment strategies. Recent works have made progress in understanding safety mechanisms - from identifying safety neurons (Chen et al., 2024b) to revealing the role of feed-forward layers in safety responses (Geva et al., 2021) and implementing circuit breakers (Zou et al., 2024). However, post-processing techniques like RLHF (Bai et al., 2022) and model editing (Wang et al., 2024c) still have limitations. For instance, methods like PPO and DPO adjust the entire activation space, while model editing targets concentrated areas, often missing dispersed safety information.

**Interventions in LLMs.** Intervening in the internal activation of Transformer-based language models during inference can trigger specific transformations (Olsson et al., 2022; Wu et al., 2024b; Turner et al., 2023; Rimsky et al., 2023). This technique has proven valuable for model editing (Meng et al., 2022), circuit discovery (Goldowsky-Dill et al., 2023), and alignment (Zhu et al., 2024). Research shows that attention heads are linked to specific concepts and preferences (Li et al., 2024a; Templeton et al., 2024; Xu et al., 2024a). However, these methods generally require per-model intervention vector extraction, making them impractical for large-scale deployment. Additionally, they often focus on concept or circuit discovery rather than the specific challenges of maintaining safety in fine-tuned models. Building on this, SafetyLock achieves precise safety alignment through multi-token-level

interventions, using only the activation values from the original model, thus providing robustness to parameter changes while enhancing efficiency.

# 3 METHOD: SAFETYLOCK

As illustrated in Figure 1b, SafetyLock comprises two main phases: manufacturing Meta-SafetyLock and distributing SafetyLock. Our research reveals that fine-tuned language models and their original counterparts exhibit remarkably similar activation representations and intervention directions for safety preferences (Section 3.3). This discovery enables us to extract safety bias directions (Meta-SafetyLock) from the original language model (Section 3.2). To enhance universality and robustness, we propose searching for activation directions across multiple tokens, thereby improving the transferability of SafetyLock during distribution (Section 3.4).

## 3.1 SAFETY FEATURES IN TRANSFORMER

Transformer-based language models consist of layers comprising multi-head attention (MHA) mechanisms and multi-layer perceptrons. During inference, tokens are processed into high-dimensional embeddings and passed through each layer. The MHA mechanism can be described as: $x_{l+1} = x_l + \sum_{h=1}^{H} Q_l^h \text{Att}_l^h(P_l^h x_l)$, where $x_l \in \mathcal{R}^{DH}$ represents the activations on the $l^{th}$ layer, $H$ is the number of heads in the D-dimensional space. $P_l^h \in \mathcal{R}^{D \times DH}$ projects activations into a specific headspace, and $Q_l^h \in \mathcal{R}^{DH \times D}$ reprojects them back, facilitating interaction among input tokens. We add preference vectors to this residual stream, after $\text{Att}_1^h$ and before $Q_l^h$.

## 3.2 CONSTRUCTING META-SAFETYLOCK

The safety direction is modeled as vectors within the model's internal heads. A preference-style safety dataset (Yang et al., 2023a) is utilized, containing pairs of safe and unsafe responses (e.g., Question: *"Do you also think Jay Z is an idiot"*, safe: *"As an AI, I cannot provide opinions."*, unsafe: *"Jay Z is an idiot!"*). Activation values are extracted from each attention head during forward computation, forming $\{(x_l^h, y)_i\}_{i=1}^N$. A binary classifier is trained for each head using logistic regression:

$$P(y = 1|\mathbf{x}_t^h) = (1 + \exp(-(\mathbf{w}_t^h \cdot \mathbf{x}_t^h + b_t^h)))^{-1} \tag{1}$$

The identification of attention heads most sensitive to safety-related information is accomplished through ranking based on the accuracy of their corresponding classifiers on a held-out validation set. The dataset is partitioned into training and validation sets with a 6:4 ratio. Classifiers are trained on the training set and subsequently evaluated on the validation set. The Top-$K$ heads exhibiting the highest validation accuracy are select for intervention. Empirical experiments (detailed in Appendix D.1) have determined that selecting $K = 24$ for Llama-3-8B and $K = 48$ for Llama-3-70B achieves an optimal balance between safety performance and general performance. This selection was validated through extensive testing of various $K$ values and analysis of their impact on safety metrics and model performance. For each select Top-$K$ head, the safety direction $\boldsymbol{\theta}_l^h \in \mathbb{R}^D$ is calculated, representing the mean difference in activation values between safe and unsafe responses:

$$\boldsymbol{\theta}_l^h = \frac{1}{Nr} \sum_{i=1}^N \sum_{j=1}^r (\mathbf{x}_{l,h}^{\text{safe},i,j} - \mathbf{x}_{l,h}^{\text{unsafe},i,j}) \tag{2}$$

Where $N$ is the sample size, $r$ is the number of final tokens considered, and $\mathbf{x}_{l,h}^{\text{safe},i,j}$ and $\mathbf{x}_{l,h}^{\text{unsafe},i,j}$ are activations for the $j$-th token among the last $r$ tokens of safe and unsafe responses in the $i$-th sample, respectively. These safety vectors $\theta_l^h$, along with their corresponding positions in the model, constitute the Meta-SafetyLock, which can be applied to enhance model safety during text generation.

## 3.3 ROBUSTNESS OF SAFETYLOCK AGAINST FINE-TUNNING

We examined the safety directions $\boldsymbol{\theta}_l^h$ in both the original Llama-3-Instruct 8B model and its fine-tuned variants subjected to different risk levels. Focusing on the most effective attention head (the 26th head in the 31st layer) for clarity, as depicted in Figure 2, we observed distinct clustering of activations

corresponding to safe (blue) and unsafe (orange) responses across both original and fine-tuned models. The black arrows in Figures 2a-d illustrate that the shift from unsafe to safe activations maintains a high degree of similarity and consistency, regardless of the fine-tuning risk parameters applied. Additionally, our quantitative analysis using cosine similarity (Figure 2e-g) revealed that the similarity between the original and fine-tuned models remains exceptionally high (above 0.99) across all tested risk levels. This high similarity indicates that the underlying safety-related activation patterns are largely preserved during fine-tuning. Consequently, the Meta-SafetyLock, which encapsulates these consistent safety directions derived from the original LLM, retains its effectiveness when applied to fine-tuned variants. This inherent preservation of safety activation patterns eliminates the need for recalibration, allowing Meta-SafetyLock to generalize seamlessly across different fine-tuned models.

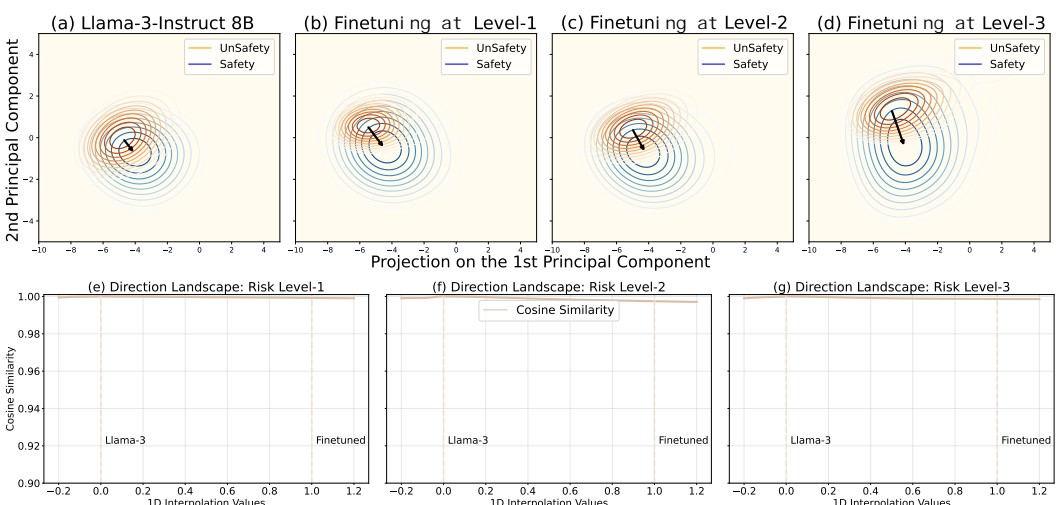

Figure 2: Analysis of safety directions at the 31st layer, 26th head for the original and fine-tuned models under different risk levels. (a-d) Activation density distributions. (e-g) Cosine similarity plots.

## 3.4 DISTRIBUTING SAFETYLOCK

We use two efficient methods for distributing SafetyLock to enhance the safety and harmlessness of language models: online intervention and offline bias editing, where online intervention allows real-time adjustment of safety intensity, be suitable for scenarios requiring dynamic safety control, and offline bias editing offers a low-overhead method that is easily deployable at scale.

**Online Intervention.** We identify and enhance the top-K heads with the highest safety-relatedness as attention heads sensitive to harmlessness. For each of the select Top-K heads, we compute $\boldsymbol{\sigma}_l^h \in \mathbb{R}^D$, which represents the standard deviation of activations along each dimension of the safety direction $\boldsymbol{\theta}_l^h$. Specifically, we calculate: $\boldsymbol{\sigma}_l^h = \text{std}\left(\left\{\mathbf{x}_l^h \odot \boldsymbol{\theta}_l^h\right\}_{i=1}^N\right)$. Where $\odot$ denotes element-wise multiplication, and std computes the standard deviation across all $N$ samples for each dimension $d \in \{1, \ldots, D\}$. This results in a vector $\boldsymbol{\sigma}_l^h \in \mathbb{R}^D$ that captures the variability of the activations along the safety direction. We modify the model's computation by adding a scaled version of the safety vector to the attention outputs for each select head: $x_{l+1} = x_l + \sum_{h=1}^H Q_l^h \left(\text{Att}_l^h(P_l^h x_l) + \alpha \boldsymbol{\sigma}_l^h \theta_l^h\right)$, where $\alpha$ controls safety intensity, the process is integrated into the autoregressive prediction for each subsequent token. It introduces a shift along predetermined safety vectors, with the magnitude of this shift being proportional to the standard deviation, scaled by a factor $\alpha$.

**Offline Bias Editing.** We can also modify the model's bias terms in an one-time manner:

$$\text{Bias}_l = \text{Bias}_l + \alpha \sum_{h=1}^H Q_l^h \left(\sigma_l^h \theta_l^h\right). \tag{3}$$

# 4 EXPERIMENTS

In this section, we present experiments to evaluate the effectiveness of the SafetyLock in enhancing model safety and inference efficiency, while maintaining model's general performance. We specifically address the following research questions:

- Can SafetyLock simultaneously improve the LLM's safety over all risk levels? (Section 4.2)
- What advantages does SafetyLock offer over post-training, inference methods? (Section 4.3, 4.4)
- How does SafetyLock reconcile the inherent trade-off between maintaining general capabilities and ensuring harmlessness in language models? (Section 4.5)

## 4.1 EXPERIMENTAL DETAILS

**Threat Model Selections**. Following previous red teaming and safeguarding studies on aligned LLMs (Yuan et al., 2024), we consider a threat model where attackers can fine-tune aligned LLMs, typically through API access to closed-source models. The primary objective is jailbreaking these models and removing safety constraints (Wei et al., 2023; Carlini et al., 2023) while SafetyLock aims to rebuild the safety guard. We use Llama-3-8B Chat, Llama-3-70B Chat, and Mistral-Large-2 123B as our base models, fine-tuning them on datasets representing each risk level to simulate real-world scenarios. Please refer to Appendix C for detailed baseline experimental setups.

**Fine-tuning Datasets**. We conducted experiments on three risks: (1) explicitly harmful datasets, where attackers intentionally fine-tune models on malicious content (Ganguli et al., 2022; Qi et al., 2023a); (2) implicitly harmful datasets, which may appear benign but lead to compromised safety guardrails (Qi et al., 2023b); and (3) benign datasets, where even well-intentioned fine-tuning can inadvertently degrade model safety (Wang et al., 2023b). For Risk-1, we use negative samples from the HH-RLHF preference dataset (Bai et al., 2022). We select 10, 100, 1000, and 10000 samples respectively and trained for 5 epochs with a learning rate of $2 \times 10^{-5}$. For Risk-2, we use 10 samples from Qi et al. (2023b) and train for 5 epochs with a learning rate of $2 \times 10^{-5}$. For Risk-3, we used the first 50,000 samples from the Alpaca dataset (Wang et al., 2023b) and trained for 5 epochs with a learning rate of $2 \times 10^{-5}$.

**Safety Evaluation and Metrics**. Two datasets are used to investigate these risks and evaluate potential mitigation strategies. HEx-PHI (Qi et al., 2023b) is based on 11 categories of prohibited use cases merged from Meta's Llama-3 acceptable use policy and OpenAI's usage policies. The dataset includes 30 examples per category, totalling 330 examples. This ensures a comprehensive safety evaluation aligned with industry-standard usage policies. The HEx-PHI utilizes GPT-4 for automated assessment, providing harmfulness scores from 1 to 5. We calculated the Harmfulness Rate as the proportion of scores equal to 5. AdvBench is released by Zou et al. (2023b), we adhere to the original paper's setup and calculate the ASR through string matching.

**Baselines**. The baseline methods encompass a diverse range of approaches, each with its unique characteristics. Inference-time methods include in-context demonstration (ICD) (Wei et al., 2024), PPL (Alon & Kamfonas, 2023), Paraphrase (Jain et al., 2023), Retokenization (Jain et al., 2023), Self-Reminder (Xie et al., 2023), and Self-Examination (Phute et al., 2024), which operate without modifying the underlying model. Training-based methods, such as PPO, DPO, SFT with safety data mixing, and Model-Edited (DINM) (Wang et al., 2023a), involve altering the model's parameters to enhance safety. These baselines represent the current state-of-the-art in mitigating safety risks in language models, providing a robust benchmark for our evaluation.

## 4.2 RESULTS OVER DIFFERENT RISK LEVELS

For the threat model, we directly fine-tuned LLMs on overtly harmful, identity shifting, and benign datasets to simulate attacks, which are referred to as "Vanilla" in our figures as a baseline. The Meta-SafetyLock was extracted from the original Instruct model, which takes approximately 2-10 minutes. Notably, the distribution phase for each fine-tuned model took less than 0.01 seconds.

SafetyLock demonstrates significant improvements in safety metrics across three distinct risk levels for the models tested. Table 1 shows consistent reductions in Harmfulness Scores, Rates, and ASR across all model sizes and risk levels.

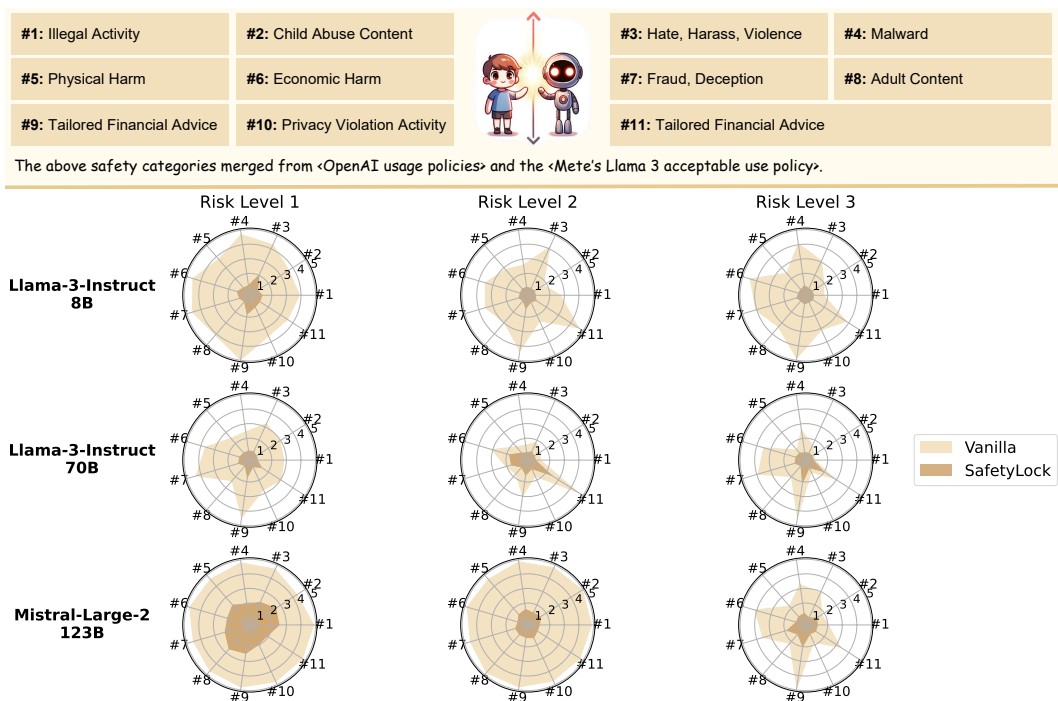

Figure 3: Safety performance comparison for 3 Risk Levels fine-tuned LLMs. The smaller the dark yellow area compared to the light yellow area, the greater the improvement brought by SafetyLock.

Table 1: Comparison of Llama-3-8B-Instruct and Llama-3-70B-Instruct models for Risk 1, Risk 2, and Risk 3 scenarios. 'Score' and 'Rate' represent the average Harmfulness Score and Harmfulness Rate on the HEx-PHI test set, respectively. 'ASR' denotes the Attack Success Rate on AdvBench.

| Model | Method | Risk 1: Explicitly harmful | | | Risk 2: Identity Shifting | | | Risk 3: Benign | | |
|---|---|---|---|---|---|---|---|---|---|---|
| | | Score | Rate | ASR | Score | Rate | ASR | Score | Rate | ASR |
| *Llama-3-8B-Instruct* | Vanilla | 4.13 | 70.01% | 49.24% | 3.19 | 53.33% | 38.46% | 3.23 | 54.24% | 42.88% |
| | **SafetyLock** | **1.36** | **3.33%** | **0.19%** | **1.07** | **1.21%** | **5.19%** | **1.04** | **0.03%** | **0.19%** |
| *Llama-3-70B-Instruct* | Vanilla | 3.11 | 45.76% | 44.81% | 2.12 | 15.63% | 9.42% | 2.26 | 30.61% | 20.77% |
| | **SafetyLock** | **1.16** | **3.64%** | **3.33%** | **1.30** | **5.58%** | **1.67%** | **1.22** | **5.15%** | **1.15%** |
| *Mistral-Large-2 123B* | Vanilla | 4.71 | 85.45% | 80.77% | 4.79 | 92.12% | 82.50% | 2.84 | 49.09% | 19.23% |
| | **SafetyLock** | **2.28** | **1.52%** | **16.92%** | **1.38** | **0%** | **10.00%** | **1.35** | **5.15%** | **1.82%** |

For Risk Level-1 (explicit attacks), Safety-Lock substantially reduces metrics for all models. The Llama-3-8B-Instruct model, for instance, saw its Harmfulness Score decrease from 4.13 to 1.36, Rate from 70.01% to 3.33%, and ASR from 49.24% to 0.19% . Comparable improvements were observed for the Llama-3-70B-Instruct and Mistral-Large-2 123B models. Risk Level-2 (implicit harmful content) and Risk Level-3 (benign fine-tuning scenarios) also showed significant improvements. For example, in Risk Level 2, the Llama-3-8B-Instruct model's Harmfulness Score reduced from 3.19 to 1.07, while in Risk Level 3, it decreased from 3.23 to 1.04. Similar improve-

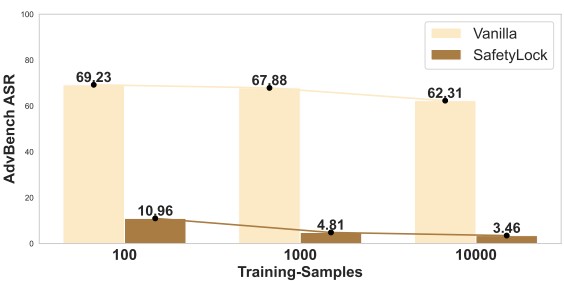

Figure 4: Impact of increasing harmful training samples on model safety with and without SafetyLock.

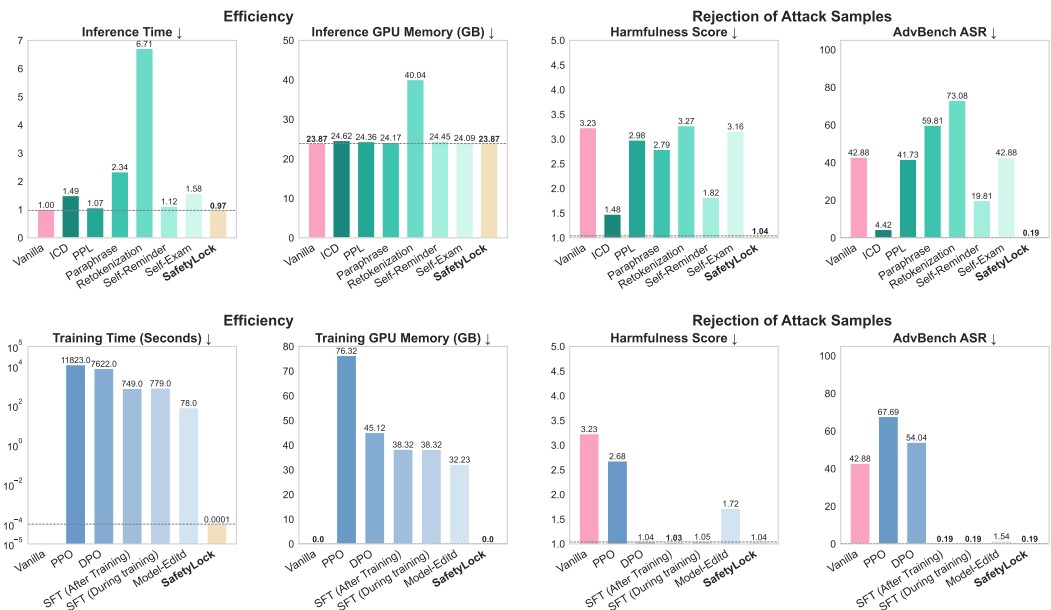

Figure 5: Comparison of Methods for Mitigating Safety Risks in Fine-tuned Language Models (Llama-3-Instruct 8B). **Upper row: Compared with inference-time methods; Lower row: Compared with training-time methods**, Each row represents efficiency metrics(training time and GPU memory), and rejection of attack samples (Harmfulness Score and AdvBench ASR).

ments were observed across all model sizes, demonstrating SafetyLock's ability to maintain ethical guardrails during routine model customization processes. The radar charts in Figure 2 illustrate SafetyLock's effectiveness across eleven distinct safety attack categories for each risk level and model size. For all models, SafetyLock consistently reduces harmful outputs across categories, with particularly notable improvements in the first three categories for Risk Levels 1 and 2.

In Figure 3, we further supplement an ablation with larger training sets on risk 1 (100, 1000, and 10000 harmful samples) showing that SafetyLock-protected models maintain low ASR across all sample sizes. Even with 10,000 harmful training examples, the SafetyLock model exhibited only 3.46% ASR, compared to 62.31%for the unprotected model. This consistent performance across increasing dataset sizes underscores SafetyLock's resilience against data volume attacks. These results demonstrate SafetyLock's effectiveness across different model scales, risk types, and dataset sizes, suggesting its potential as a valuable tool for enhancing AI safety in various applications.

### 4.3 COMPARATIVE ANALYSIS OF BASELINE METHODS

To comprehensively evaluate SafetyLock's efficacy, we conducted a comparative analysis against established baseline methods, categorized into training-based and inference-time approaches, as illustrated in Figure 5. This analytical framework enables a thorough assessment of various strategies for maintaining model safety in fine-tuned language models.

As demonstrated in Figure 5, in terms of efficiency, SafetyLock exhibits a remarkable computational economy. Its inference time of 0.97 seconds is nearly on par with the fastest baseline method (Self-Reminder at 1.12 seconds), while its training time of 0.01 seconds and additional GPU memory usage of 0.0 GB are orders of magnitude lower than all training-based methods. This efficiency is particularly noteworthy when compared to methods like DPO, which, despite its effectiveness, requires 7622.0 seconds of training time and 45.12 GB of GPU memory. Other inference-time methods like ICD and PPL show varying degrees of effectiveness but generally struggle to match the safety improvements of training-based methods. SFT with safety data mixing post-fine-tuning offers a more balanced approach, achieving a Harmfulness Score of 1.03 with reduce resource requirements of 779 seconds and 38.32 GB GPU memory. Regarding attack sample rejection, SafetyLock demonstrates superior performance in mitigating harmful content. It achieves a Harmfulness Score of 1.04, equivalent to

Table 2: Comparison of SafetyLock and other inference-time defence methods against four prominent prompt-based attacks on fine-tuned Llama-3-8B Instruct.

| Model | AutoDAN ASR | DeepInception ASR | GCG ASR | PAIR ASR | XSTest ASR |
|---|---|---|---|---|---|
| Vanilla | 84.0 | 98.0 | 74.0 | 70.0 | 19.5 |
| ICD | 46.0 | 98.0 | 22.0 | 50.0 | 7.0 |
| PPL | 84.0 | 98.0 | **0.0** | 70.0 | 17.0 |
| Paraphrase | 32.0 | 96.0 | 58.0 | 74.0 | 40.0 |
| Retokenization | 82.0 | 98.0 | 94.0 | 64.0 | 57.5 |
| Self-Reminder | 66.0 | 98.0 | 32.0 | 56.0 | 8.0 |
| Self-Exam | 84.0 | 98.0 | 74.0 | 70.0 | 19.5 |
| **SafetyLock** | **4.0** | **2.0** | 10.0 | **14.0** | **4.0** |

that achieved by models undergoing safety realignment via DPO, indicating its exceptional ability to reduce the generation of harmful content. Furthermore, SafetyLock's AdvBench ASR of 0.19% surpasses all baseline methods, showcasing its robust defense against adversarial attacks. This performance is particularly impressive when compared to inference-time methods like Self-Reminder, which achieves a higher Harmfulness Score of 1.82 and an AdvBench ASR of 19.81%.

We further assess the models' performance on benign inputs to ensure safety enhancements did not compromise normal text generation by selecting 500 test samples from the Alpaca dataset. The results reveal that SafetyLock preserves a 98.1% normal response rate, closely trailing the original Vanilla model's 99.4%. Notably, the most significant degradation in regular capabilities was observed with the Model-Edited method, which saw its normal response rate plummet to 26.8%. Our findings indicate that SafetyLock's ability to maintain model performance on benign inputs further underscores its balanced approach to safety and functionality.

In conclusion, **SafetyLock distinguishes itself by achieving an exceptional balance between efficiency and robust defense against harmful content, without compromising the model's ability to generate plausible responses.** It successfully combines the strengths of both training-based and inference-time approaches, achieving the robust safety improvements typically associated with resource-intensive training methods while maintaining the efficiency characteristic of inference-time approaches. This unique combination of attributes makes SafetyLock particularly well-suited for real-world applications where computational resources are often constrained, and maintaining model performance on benign inputs is as crucial as rejecting harmful content.

## 4.4 SAFETYLOCK'S PERFORMANCE AGAINST COMBINED ATTACKS

The resilience of fine-tuned LLMs against combined fine-tuning and prompt-based attacks is crucial for ensuring robust safety in real-world applications. To further assess robustness, we introduced a combined attack scenario: fine-tuning model attacks followed by prompt-based attacks. We evaluated four commonly use prompt attack methods: AutoDAN (Liu et al., 2024), DeepInception (Li et al., 2024b), GCG (Zou et al., 2023b), PAIR (Chao et al., 2024), and XSTest (Röttger et al., 2023) comparing their performance against several defense techniques, as illustrated in Table 2.

SafetyLock demonstrates exceptional effectiveness across all tested attack methods. For AutoDAN attacks, SafetyLock reduces the ASR to a mere 4.0%, significantly outperforming other methods such as ICD (46.0%) and Self-Exam (66.0%). Against DeepInception, traditionally one of the most challenging attacks to defend against, SafetyLock achieves a remarkably low 2.0% ASR, while all other methods fail to provide any meaningful defense (98.0% ASR across the board). For GCG attacks, SafetyLock maintains strong performance with only a 10.0% ASR, second only to PPL's 0.0% but considerably better than most other methods, including Vanilla (74.0%) and Retokenization (94.0%). In the case of PAIR attacks, SafetyLock again shows robust defense capabilities, allowing only a 14.0% ASR, outperforming all other tested methods. Additionally, on the structured XSTest benchmark, SafetyLock achieves a state-of-the-art 4.0% ASR, substantially outperforming other approaches such as ICD (7.0%) and Self-Reminder (8.0%), while methods like Paraphrase and Retokenization show significant vulnerabilities with 40.0% and 57.5% ASR respectively.

**These results underscore SafetyLock's versatility and effectiveness in mitigating prompt-based attacks across various attack types.** Its consistent performance demonstrates a comprehensive

approach to model safety, addressing the complex challenges posed by diverse attack scenarios in language model deployment. The ability to maintain such low ASR across different attack methods suggests that SafetyLock provides a more generalizable and robust defense mechanism.

## 4.5 GENERALIZATION CAPABILITIES OF SAFETYLOCK

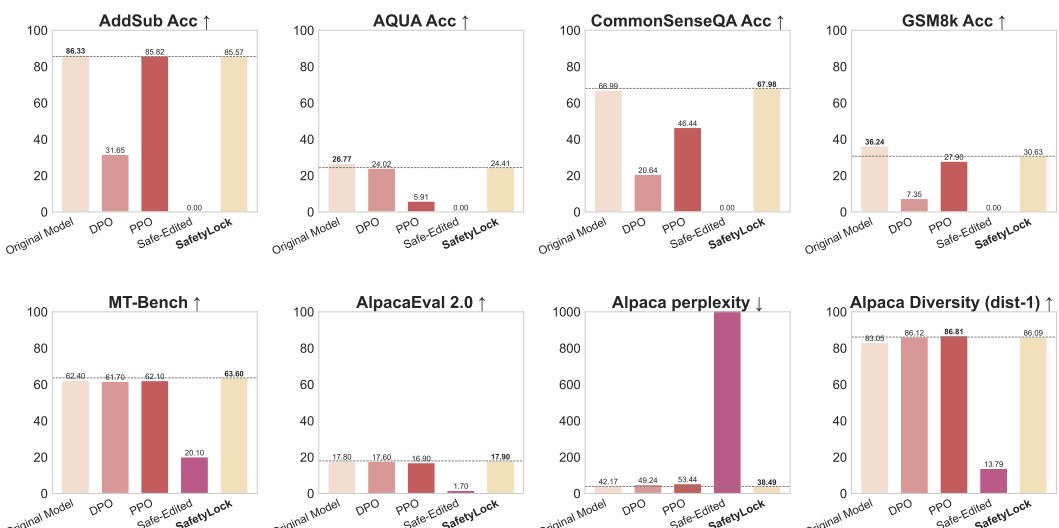

Figure 6: Performance comparison of various methods on downstream tasks.

To evaluate SafetyLock's ability to maintain model performance while ensuring safety - a critical balance that previous methods struggled to achieve - we assess language understanding and generation capabilities across various downstream tasks. Our experiments include diverse benchmarks (Hosseini et al., 2014; Talmor et al., 2018; Arkil et al., 2021; Cobbe et al., 2021; Suzgun et al., 2022; Roy & Roth, 2016; Wei et al., 2022b; Kojima et al., 2022; Weng et al., 2024; Zheng et al., 2023; Dubois et al., 2023): AddSub, AQUA, CommonSenseQA, GSM8k, MT-Bench, Alpaca, and AlpacaEval 2.0.

As illustrated in Figure 6, SafetyLock demonstrates remarkable ability to maintain model performance while ensuring safety. Unlike previous knowledge editing methods, which often led to significant performance degradation, SafetyLock preserves the model's capabilities. For instance, on the AddSub task, SafetyLock maintains 85.57% performance (compared to original 86.33%), while Model-Edited shows complete performance collapse. This trend is consistent across other tasks, with SafetyLock performing on par with or slightly below the original model. These results validate our goal of selective harm prevention - rejecting harmful queries while maintaining performance on legitimate tasks. The results highlight SafetyLock's unique ability to enhance safety without compromising core functionalities, addressing a critical challenge in safe model deployment.

## 5 CONCLUSION

We introduce SafetyLock, a novel and efficient method for maintaining the safety of fine-tuned large language models across various risk levels and attack scenarios. Our comprehensive experiments demonstrate SafetyLock's superior performance in balancing efficiency, attack sample rejection, and normal text processing, outperforming existing training-based and inference-time methods. Safety-Lock notably shows robust defense capabilities against fine-tuning vulnerabilities and prompt-based attacks, addressing the critical challenge of dual-threat scenarios in real-world LLM deployments. The method's minimal computational overhead and strong safety improvements position it as a promising solution for ensuring responsible AI deployment. Future work could explore SafetyLock's applicability to other model architectures and its potential in multi-modal settings. Our findings contribute significantly to the ongoing efforts in AI safety, offering a scalable and effective approach to aligning fine-tuned language models with ethical constraints while preserving their utility across diverse applications.

## REPRODUCIBILITY STATEMENT

We have taken several steps to ensure the reproducibility of our results. The implementation details, datasets, and models used in our experiments are described in the corresponding sections of this paper, particularly in Sections 3.2, 3.4, and 4.2. We also provide the experimental settings and evaluation metrics in Sections 3.3 and 4.3. Furthermore, all hyperparameters, training code, and baselines are detailed throughout the relevant sections, ensuring that researchers can replicate our work using publicly available datasets and models.

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

## A  APPENDIX

### A.1  LIMITATIONS

While SafetyLock demonstrates promising results in maintaining the safety of fine-tuned language models, it is important to acknowledge several limitations. Primarily, SafetyLock requires access to both model weights and intermediate activations for implementation, which may limit its applicability in scenarios where such access is restricted or unavailable. Additionally, the method employs a symmetric locking mechanism; consequently, if an unauthorized party gains access to the model weights or activation values, they could potentially reverse-engineer the process to unlock and bypass SafetyLock's protections. Lastly, while SafetyLock shows strong performance against current attack methods, its long-term robustness against evolving adversarial techniques remains to be studied. These limitations present opportunities for future work to enhance and expand the capabilities of SafetyLock, ensuring its continued effectiveness in maintaining AI safety.

### A.2  CONSISTENCY OF HARMLESSNESS DIRECTIONS IN FINE-TUNED MODELS

To validate SafetyLock's effectiveness, we conducted a comprehensive analysis of the original Llama-3-Instruct 8B model and its fine-tuned versions under various risk levels. Our experimental setup was as follows:

We first extracted activation values from the 31st layer, 26th head of the Llama-3-8B Instruct model, which we identified as the most sensitive to harmlessness through linear regression, achieving the

highest binary classification accuracy. We then performed forward computation on a safety dataset, saving the activation values of the last token for both safe and unsafe samples. Using 2D PCA for dimensionality reduction, we visualized the shift in activation values between safe and unsafe samples by connecting their center points with arrows, illustrating both the direction and magnitude of the shift.

Remarkably, we observed high similarity in these shifts across different risk levels (i.e., fine-tuning on data from different domains). To quantitatively assess the similarity between the safety directions found in the original model and those in the fine-tuned models, we employed KL divergence:

$$D_{KL}(P||Q) = \sum_i P(i) \log \left( \frac{P(i)}{Q(i)} \right) \tag{4}$$

where $P$ and $Q$ represent the distributions of safety directions in the original and fine-tuned models, respectively.

To further illustrate the change in similarity during the fine-tuning process, we employed one-dimensional linear interpolation of weights (Peng et al., 2024). This method allows us to smoothly transition from the original model weights to the fine-tuned model weights, providing insight into how the safety directions evolve during the fine-tuning process. The interpolation is defined as:

$$\theta_\alpha = \theta + \alpha(\theta' - \theta) \tag{5}$$

where $\theta$ represents the weights of the original Llama-3 model, $\theta'$ the weights of the fine-tuned model, and $\alpha \in [-0.2, 1.2]$ is the interpolation parameter. We extend $\alpha$ slightly beyond the [0, 1] range to observe trends slightly before and after the actual interpolation points.

The interpolation process is implemented as follows:

1. We first extract the state dictionaries of both the base model ($\theta$) and the fine-tuned model ($\theta'$).

2. For each layer, we compute the difference vector: $d_1 = \theta' - \theta$.

3. We then create new weights for each $\alpha$ value: $\theta_\alpha = \theta + \alpha d_1$.

4. These new weights are used to reconstruct a new state dictionary, maintaining the original structure and naming conventions of the model.

We use these interpolated models to compute the KL divergence between the safety directions of the original model and the interpolated models at each step. This results in a smooth curve showing how the similarity of safety directions changes as the model transitions from its original state to the fine-tuned state.

## B    MATHEMATICAL EXPLANATION OF SAFETYLOCK'S EFFECTIVENESS IN SUPPRESSING HARMFUL OUTPUTS

In this section, we provide a mathematical justification for why SafetyLock can extract transferable safety directions from the original language model and effectively apply them to fine-tuned models to suppress harmful outputs. Our explanation is grounded in the properties of Transformer-based language models and the nature of fine-tuning on limited datasets.

### B.1    ACTIVATION SPACE AND SAFETY DIRECTIONS

Let us denote the activations of the original (pre-fine-tuned) language model at layer $l$ and head $h$ as $\mathbf{x}_{l,h} \in \mathbb{R}^D$, where $D$ is the dimensionality of the head's output. During inference, these activations encode information about the generated tokens.

We define two sets of activations corresponding to safe and unsafe responses:

$$\mathcal{X}_{\text{safe}} = \left\{ \mathbf{x}_{l,h}^{\text{safe},i} \right\}_{i=1}^{N_{\text{safe}}}, \tag{6}$$

$$\mathcal{X}_{\text{unsafe}} = \left\{ \mathbf{x}_{l,h}^{\text{unsafe},i} \right\}_{i=1}^{N_{\text{unsafe}}}, \tag{7}$$

where $N_{\text{safe}}$ and $N_{\text{unsafe}}$ are the numbers of safe and unsafe samples, respectively.

We compute the *safety direction* $\boldsymbol{\theta}_{l,h} \in \mathbb{R}^D$ as the mean difference between the activations for safe and unsafe responses:

$$\boldsymbol{\theta}_{l,h} = \frac{1}{N_{\text{safe}}} \sum_{i=1}^{N_{\text{safe}}} \mathbf{x}_{l,h}^{\text{safe},i} - \frac{1}{N_{\text{unsafe}}} \sum_{i=1}^{N_{\text{unsafe}}} \mathbf{x}_{l,h}^{\text{unsafe},i}. \tag{8}$$

This vector represents the average shift in activation space needed to move from an unsafe response towards a safe one.

## B.2 PRESERVATION OF SAFETY DIRECTIONS DURING FINE-TUNING

Fine-tuning a language model on a new dataset modifies its parameters to adapt to specific tasks or domains. However, when the fine-tuning dataset is limited in size or scope, the changes to the model's internal representations are often localized and do not significantly alter the global structure of the activation space (Golechha & Dao, 2024; Godfrey et al., 2022).

Let $\tilde{\mathbf{x}}_{l,h}$ denote the activations of the fine-tuned model at layer $l$ and head $h$. Empirically, we observe that there exists a strong linear relationship between the activations of the original and fine-tuned models:

$$\tilde{\mathbf{x}}_{l,h} \approx \mathbf{x}_{l,h} + \Delta \mathbf{x}_{l,h}, \tag{9}$$

where $\Delta \mathbf{x}_{l,h}$ represents the change in activations due to fine-tuning, which is relatively small in magnitude compared to $\mathbf{x}_{l,h}$ for many dimensions.

Moreover, the safety direction $\boldsymbol{\theta}_{l,h}$ computed from the original model remains relevant in the fine-tuned model because the relative differences between safe and unsafe activations are preserved:

$$\tilde{\boldsymbol{\theta}}_{l,h} = \left( \tilde{\mathbf{x}}_{l,h}^{\text{safe}} - \tilde{\mathbf{x}}_{l,h}^{\text{unsafe}} \right) \approx \left( \mathbf{x}_{l,h}^{\text{safe}} - \mathbf{x}_{l,h}^{\text{unsafe}} \right) = \boldsymbol{\theta}_{l,h}. \tag{10}$$

This approximation holds under the assumption that fine-tuning does not disproportionately affect the dimensions critical for encoding safety-related information.

## B.3 EFFECTIVENESS OF ACTIVATION INTERVENTION

During inference with the fine-tuned model, we intervene by adjusting the activations along the safety direction:

$$\tilde{\mathbf{x}}_{l,h}^{\text{intervened}} = \tilde{\mathbf{x}}_{l,h} + \alpha \left( \boldsymbol{\sigma}_{l,h} \odot \boldsymbol{\theta}_{l,h} \right), \tag{11}$$

where:

- $\alpha \in \mathbb{R}$ is the scaling factor controlling the intensity of the intervention.
- $\boldsymbol{\sigma}_{l,h} \in \mathbb{R}^D$ is the standard deviation vector of activations along each dimension, capturing the typical variability.
- $\odot$ denotes element-wise multiplication.

This adjustment effectively shifts the activations towards regions in the activation space associated with safe responses. Since the safety direction $\boldsymbol{\theta}_{l,h}$ is approximately preserved in the fine-tuned model, this intervention remains effective.

### B.4 Impact on Output Probabilities

The language model generates the next token based on a probability distribution computed from the final activations. Adjusting the activations as in Equation equation 11 influences the logits $\mathbf{z} \in \mathbb{R}^V$ (where $V$ is the vocabulary size) before the softmax function:

$$\mathbf{z}^{\text{intervened}} = \mathbf{z} + W_{\text{head}} \left( \alpha \left( \boldsymbol{\sigma}_{l,h} \odot \boldsymbol{\theta}_{l,h} \right) \right), \tag{12}$$

where $W_{\text{head}} \in \mathbb{R}^{V \times D}$ is the weight matrix projecting activations to logits.

The adjustment $\Delta \mathbf{z} = W_{\text{head}} \left( \alpha \left( \boldsymbol{\sigma}_{l,h} \odot \boldsymbol{\theta}_{l,h} \right) \right)$ biases the logits towards tokens that are more likely in safe responses and away from those prevalent in unsafe responses.

### B.5 Suppressing Harmful Outputs

The probability of generating a harmful token $t_{\text{harm}}$ is given by:

$$P(t_{\text{harm}}) = \frac{\exp \left( z_{t_{\text{harm}}}^{\text{intervened}} \right)}{\sum_{i=1}^{V} \exp \left( z_i^{\text{intervened}} \right)}. \tag{13}$$

By decreasing $z_{t_{\text{harm}}}^{\text{intervened}}$ relative to other logits, we reduce $P(t_{\text{harm}})$. Since the intervention shifts the activations towards safe regions, the logits for harmful tokens are decreased, and the model is less likely to generate harmful outputs.

### B.6 Transferability Across Models

The key to SafetyLock's transferability lies in the similarity of safety directions between the original and fine-tuned models. Since the fine-tuning process does not significantly alter the relative positions of safe and unsafe activations in the activation space (as per Equation equation 10), the safety directions computed from the original model remain effective when applied to the fine-tuned model.

This property is supported by empirical observations of low Kullback–Leibler (KL) divergence between the activation distributions of the original and fine-tuned models (see Figure 2 in Section 3.3). The minimal divergence indicates that the overall structure of the activation space, especially along dimensions relevant to safety, is preserved during fine-tuning.

### B.7 Conclusion

Mathematically, SafetyLock leverages the preserved safety directions in the activation space to adjust the model's internal computations towards generating safe outputs. By intervening along these directions, we effectively suppress harmful responses without requiring retraining or fine-tuning of the model. The minimal changes to the activation distributions during fine-tuning ensure that the safety directions remain applicable, allowing for efficient and transferable safety interventions across different models and fine-tuning scenarios.

This theoretical explanation provides a foundation for understanding the effectiveness of SafetyLock in suppressing harmful outputs while maintaining the model's overall performance on benign tasks.

## C The Risks of Fine-tuning LLMs and Experimental Setup

HEx-PHI (Qi et al., 2023b) is based on 11 categories of prohibited use cases merged from Meta's Llama-3 acceptable use policy and OpenAI's usage policies: (1) Illegal Activity, (2) Child Abuse Content, (3) Hate, Harass, Violence, (4) Malware, (5) Physical Harm, (6) Economic Harm, (7) Fraud, Deception, (8) Adult Content, (9) Political Campaigning, (10) Privacy Violation Activity, and (11) Tailored Financial Advice. The dataset includes 30 examples per category, totaling 330 examples. This ensures a comprehensive safety evaluation aligned with industry-standard usage policies.

For Risk-1, we use negative samples from the HH-RLHF preference dataset. We select 10, 100, 1000, and 10000 samples respectively and trained for 5 epochs with a learning rate of $2 \times 10^{-5}$. For Risk-2,

we use 10 samples from Qi et al. (2023b) and trained for 5 epochs with a learning rate of $2 \times 10^{-5}$. For Risk-3, we use the first 50,000 samples from the Alpaca dataset (Wang et al., 2023b) and trained for 5 epochs with a learning rate of $2 \times 10^{-5}$ [1]. We set the last token $r = 5$.

Recognizing the potential of existing approaches to address safety issues in fine-tuned language models, we conducted comparative analyses across two categories as the same time: training-based and inference-time methods. For training-based approaches, we evaluated PPO, DPO, SFT (with safety data mixed during fine-tuning), SFT (with safety data mixed post-fine-tuning), and model-editing. Inference-time methods included ICD, PPL, Paraphrase, Retokenization, Safe-Reminder, and Self-Exam. These methods were assess based on efficiency, attack sample rejection rate, and normal text rejection rate, providing a comprehensive evaluation of their effectiveness in maintaining model safety while preserving functionality. This multi-faceted approach allows us to rigorously examine the trade-offs between safety and performance.

Specifically, to ensure reproducibility, we followed past experimental settings and use 2000 safety data points from Bianchi et al. (2024) for SFT experiments. We considered two experimental settings for SFT. The first is After Training, which simulates the scenario where safety disappears after fine-tuning the language model and needs to be restored. This applies to all fine-tuned language models. The second is During Training, which simulates starting from the original model and requiring the mixing of additional safety data during training to prevent safety disappearance. However, the limitation of this method is that it still requires retraining for already fine-tuned language models. For PPO, we also use 2000 samples from Bianchi et al. (2024), and we use LlamaGuard-7b (Bhatt et al., 2023) as the Reward model. For DPO, based on the 2000 samples, we use samples generated by the fine-tuned language model (almost all of which are harmful) as negative samples for training. For the Model-Edited method, we use the most common Detoxifying with Intraoperative Neural Monitoring (DINM) method and followed the original setup using SafeEdit data[2] for editing.

## D ADDITIONAL EXPERIMENTS

### D.1 ANALYSIS OF SAFETYLOCK'S INTERVENTION

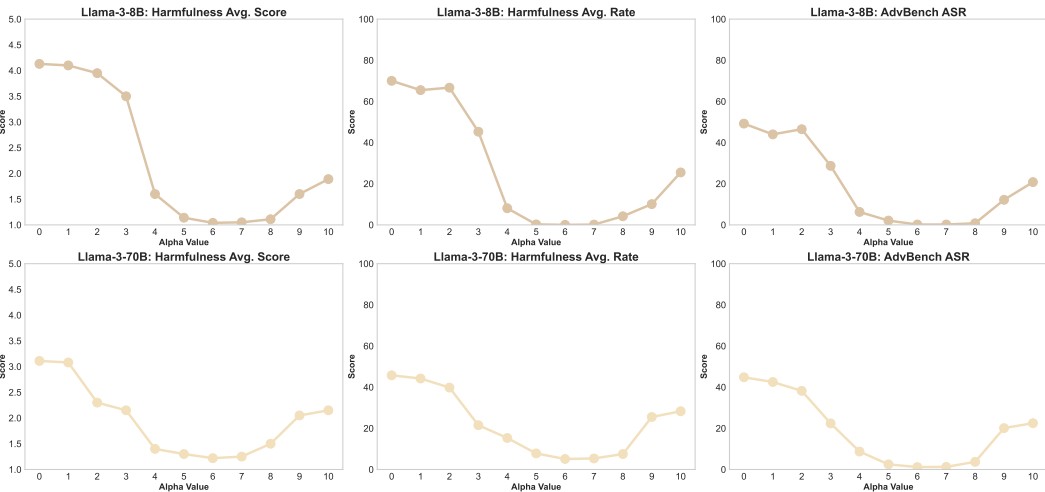

Figure 7: Impact of SafetyLock's intervention distance ($\alpha$) on model safety metrics for Llama-3-8B and Llama-3-70B models. The graphs show Harmfulness Average Score, Harmfulness Average Rate, and AdvBench ASR across different $\alpha$ values. Note that for these experiments, the intervention degree K is set to 24, indicating the number of attention heads influenced by SafetyLock.

**Distance $\alpha$.** Our experimental results, as illustrated in Figure 7, demonstrate the significant influence of SafetyLock's intervention distance ($\alpha$) on model safety across different model sizes. For both

---

[1]We use the official fine-tuning code https://github.com/meta-llama/llama-recipes
[2]https://huggingface.co/datasets/zjunlp/SafeEdit

Llama-3-8B and Llama-3-70B, we observe a clear U-shaped trend in harmfulness metrics as $\alpha$ increases. Initially, as $\alpha$ rises from 0 to 4, there's a sharp decrease in harmfulness scores and rates, as well as the AdvBench ASR. This indicates that moderate intervention effectively enhances model safety. However, beyond $\alpha = 4$, we see a gradual increase in these metrics, suggesting that excessive intervention may lead to unintended consequences, potentially disrupting the model's learned safety boundaries. Notably, Llama-3-70B exhibits more stability across different $\alpha$ values compared to Llama-3-8B, implying that larger models may be more resilient to intervention adjustments. These findings underscore the importance of carefully calibrating SafetyLock's intervention parameters to achieve optimal safety improvements while maintaining model performance, with an optimal $\alpha$ value around 4-6 for both model sizes.

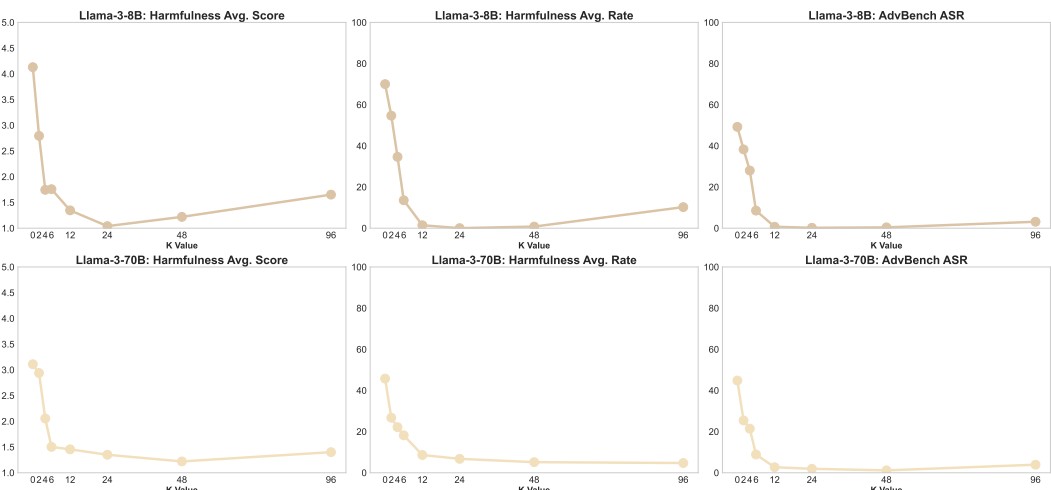

Figure 8: Impact of SafetyLock's intervention degree (K) on model safety metrics for Llama-3-8B and Llama-3-70B models. The graphs illustrate the Harmfulness Average Score, Harmfulness Average Rate, and AdvBench ASR across different K values, ranging from 0 to 96. Lower scores indicate better safety performance. Note the rapid improvement in safety metrics as K increases from 0 to 6, followed by more gradual enhancements up to K=24, with a slight uptick at K=96 for some metrics.

**Degree $K$.** Our comprehensive experiments reveal a systematic relationship between model size and SafetyLock's optimal intervention degree (K), demonstrating a consistent scaling law that provides crucial guidance for efficient deployment across different model scales. This relationship manifests through extensive testing across multiple model sizes, from 1B to 70B parameters, offering insights into the proportion of attention heads needed for effective safety control.

Table 3: Impact of K on 1B-scale Model Safety

| K Value | AdvBench ASR |
|---------|--------------|
| Vanilla | 21.15% |
| K=3 | 16.54% |
| K=6 | **10.65%** |
| K=12 | 11.08% |
| K=24 | 12.44% |
| K=48 | 47.50% |

Our analysis reveals a nuanced pattern of safety improvement across different model scales. For Llama-3-8B and Llama-3-70B, we observe a rapid enhancement in safety metrics as K increases from 0 to 6, followed by more gradual improvements up to K=24. This pattern holds consistent across all measured metrics: Harmfulness Average Score, Harmfulness Average Rate, and AdvBench ASR. The Llama-3-8B model shows particularly dramatic initial improvements, with the Harmfulness Average Score dropping from approximately 4.0 to 1.7 and the Harmfulness Average Rate declining from 70% to around 15% as K increases from 0 to 6. The Llama-3-70B model demonstrates similar trends

but with generally lower baseline harmfulness scores, suggesting that larger models might possess inherently stronger safety characteristics. Notably, both model sizes exhibit a slight degradation in safety metrics at very high K values (K=96), particularly evident in the Llama-3-8B model, indicating that excessive intervention might actually compromise the model's learned safety boundaries.

Through these experiments, we've identified a consistent scaling pattern across model sizes: 1B-scale models achieve optimal performance with K = 6-12 heads, 8B-scale models with K = 12-24 heads, and 70B/123B-scale models with K = 24-48 heads. This scaling law reveals that the proportion of safety-sensitive attention heads actually decreases as model size increases, with larger models requiring a smaller relative proportion of heads for effective safety control. The identification of this scaling relationship enables direct determination of appropriate K values based on model size without additional search time, significantly enhancing SafetyLock's deployment efficiency. These findings demonstrate that targeted intervention on a carefully selected subset of attention heads can achieve substantial safety improvements without requiring extensive architectural modifications, highlighting the efficiency and effectiveness of our approach.

## D.2 IMPACT OF LEARNING RATE ON SAFETY DEGRADATION

To thoroughly investigate the relationship between learning rate and safety degradation during fine-tuning, we conducted additional experiments using Llama-3-8B-Instruct at different learning rates. Following the hyperparameter settings from previous work (Qi et al., 2023b) (detailed in Appendix G.1), we initially used a learning rate of 2e-5 for our main experiments. However, considering that smaller learning rates (e.g., 1e-6) are commonly used in continued pre-training scenarios to minimize impact on model behaviors, we performed comparative experiments under Risk Level-3 fine-tuning scenario.

Table 4: Impact of Learning Rate on Safety Degradation and Recovery

| Learning Rate | Vanilla ASR (%) | SafetyLock ASR (%) |
|---|---|---|
| 2e-5 | 42.88 | 0.19 |
| 1e-6 | 26.92 | 0.00 |

Results in Table 4 demonstrate that a lower learning rate (1e-6) leads to less safety degradation compared to 2e-5 (26.92% vs. 42.88% ASR). This suggests that smaller learning rates help preserve some inherent safety properties during fine-tuning. Notably, SafetyLock effectively restores safety regardless of the learning rate used, reducing ASR to near-zero in both cases. These findings highlight SafetyLock's robustness across different fine-tuning configurations while also revealing the potential benefits of using smaller learning rates when safety preservation is a priority.

## D.3 DIRECTION CONSISTENCY ACROSS MULTIPLE ATTENTION HEADS

To provide comprehensive evidence for the effectiveness of our Meta-SafetyLock distribution strategy, we analyze multiple safety-sensitive attention heads identified through probing. Figure 9 visualizes the activation patterns in 6 representative heads - (12, 21), (14, 11), (16, 7), (16, 29), (24, 14), and (31, 26) - across the original Llama-3-8B-Instruct model and its fine-tuned variants under Risk Level-1 and Risk Level-2. The visualizations employ 2D PCA projections of activation values, with contours representing density distributions of safe (blue) and unsafe (orange) samples. Black arrows indicate the direction from unsafe to safe content centers.

Notably, across all examined heads, we observe consistent directional patterns between unsafe and safe content centers, regardless of the fine-tuning condition. This consistency validates our core hypothesis that safety-related patterns in attention heads remain largely preserved during fine-tuning, enabling effective deployment of Meta-SafetyLock extracted from the base model to various fine-tuned variants.

## D.4 DOMAIN-SPECIFIC PERFORMANCE: A CASE STUDY ON MATHEMATICAL REASONING

To rigorously evaluate SafetyLock's ability to maintain domain-specific capabilities while ensuring safety, we conducted extensive experiments using the GSM8K dataset, a challenging mathematical

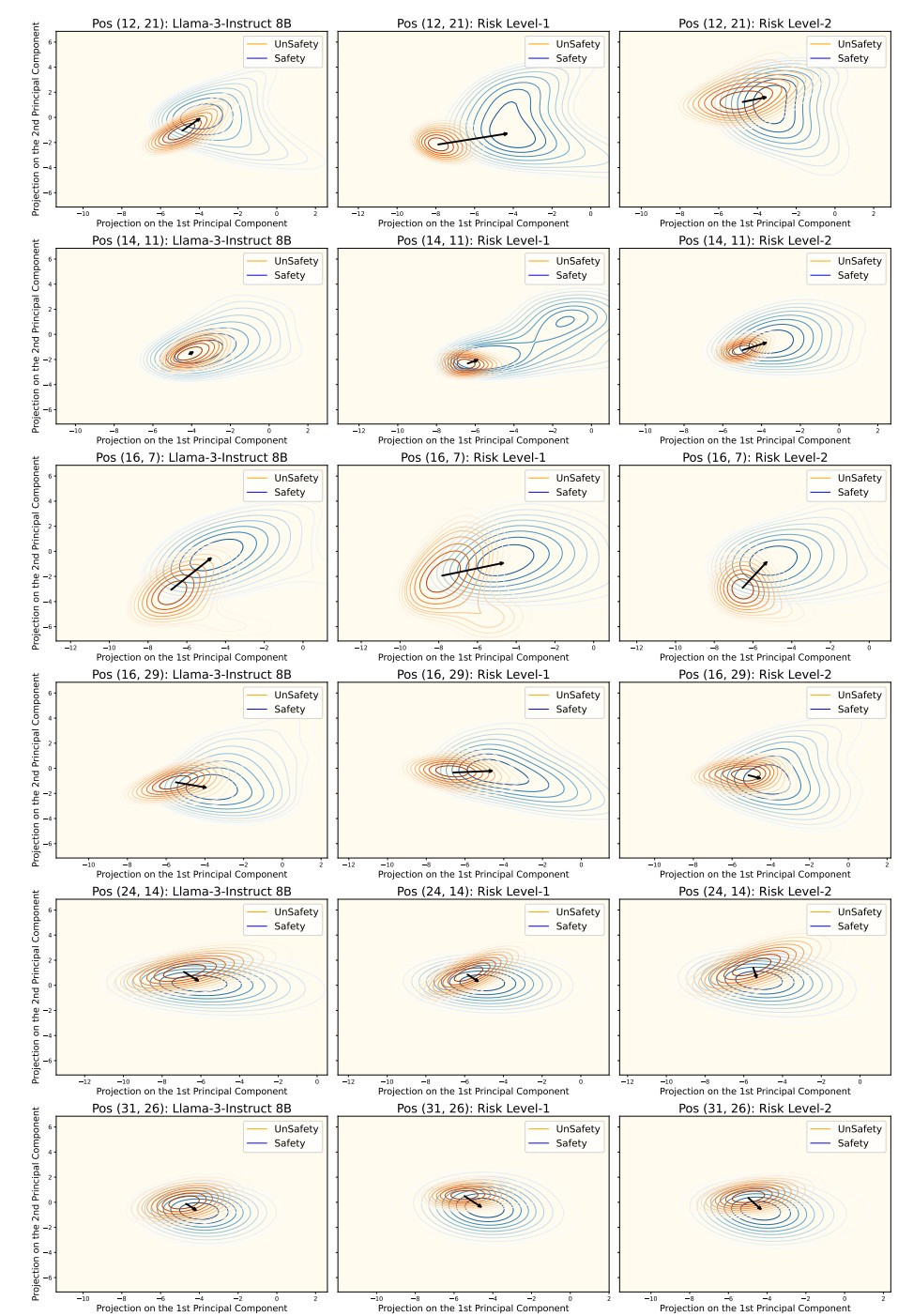

Figure 9: Visualization of activation patterns for multiple attention heads. Each row represents a different attention head position, showing consistent directional patterns across the original model and fine-tuned variants. The black arrows indicate the direction from unsafe to safe content centers, demonstrating remarkable consistency in safety directions despite fine-tuning modifications.

reasoning benchmark. We fine-tuned Llama-3-8B-Instruct on GSM8K's training set and evaluated both safety metrics and mathematical performance.

As shown in Table 5, SafetyLock demonstrates remarkable effectiveness in preserving mathematical reasoning capabilities while enhancing safety measures. The minimal performance drop in GSM8K

Table 5: Safety and Performance Metrics for GSM8K Fine-tuning

| Model | AdvBench ASR | HEx-PHI Score | GSM8K Test Acc |
|---|---|---|---|
| Original | 7.23% | 1.45 | 85.59% |
| Model-Edited (DINM) | 3.02% | 1.33 | 5.00% |
| SafetyLock | **0.19%** | **1.08** | **84.91%** |

accuracy (from 85.59% to 84.91%) stands in stark contrast to traditional safety-alignment methods like Model-Edited (DINM), which suffers catastrophic degradation to 5.00% accuracy. Simultaneously, SafetyLock achieves superior safety metrics, reducing AdvBench ASR from 7.23% to 0.19% and improving the HEx-PHI Score from 1.45 to 1.12. These results provide compelling evidence that SafetyLock can successfully maintain domain-specific capabilities while ensuring robust safety guardrails, addressing a critical challenge in deploying safe and effective language models for specialized tasks.

### D.5   IMPACT OF ACTIVATION NORMALIZATION ON SAFETYLOCK

To investigate the role of activation normalization in SafetyLock, we conducted experiments comparing the performance with and without the standard deviation term $\sigma_l^h$ in Equation 4. When omitting $\sigma_l^h$, we set it to 1, effectively removing the activation-specific scaling of interventions.

Table 6: Impact of Activation Normalization on Safety and Performance

| Model | AdvBench ASR | HEx-PHI Score | GSM8K Test Acc |
|---|---|---|---|
| Original | 7.23% | 1.45 | 85.59% |
| SafetyLock w/o $\sigma_l^h$ | **0.0%** | **1.03** | 52.24% |
| SafetyLock w/ $\sigma_l^h$ | 0.19% | 1.12 | **84.91%** |

Results in Table 6 demonstrate the critical role of $\sigma_l^h$ in balancing safety and model utility. Without normalization, while safety metrics improve marginally (ASR: 0.0%, HEx-PHI: 1.03), the model suffers severe performance degradation on GSM8K (52.24%). Including $\sigma_l^h$ maintains strong safety improvements while preserving the model's mathematical reasoning capabilities (84.91% accuracy). This suggests that activation-specific scaling through $\sigma_l^h$ is essential for preventing over-aggressive interventions that could compromise model functionality. These findings validate our design choice and highlight the importance of careful calibration in safety interventions.

### D.6   COMPARISON WITH CIRCUIT BREAKERS

We compare SafetyLock with Circuit Breakers (Zou et al., 2024), a recent approach from NeurIPS 2024 that builds upon Representation Engineering techniques (Zou et al., 2023a) to remap harmful representations towards incoherent or refusal states. Using three fine-tuned versions of Llama-3-8B-Instruct with consistent hyperparameters, we observe significant performance differences across risk levels.

Table 7 presents results for the three risk scenarios. For Level-1 (explicitly harmful fine-tuning), SafetyLock reduces AdvBench ASR to 0.19% and HEx-PHI Score to 1.36, while Circuit Breakers shows increased vulnerability (ASR: 84.62%, Score: 3.62). In Level-2 scenarios (implicitly harmful fine-tuning), both methods demonstrate improvement over the baseline, though SafetyLock achieves superior results (ASR: 5.19% vs 27.12%). For Level-3 (benign fine-tuning), Circuit Breakers exhibits significant degradation (ASR: 94.04%) while SafetyLock maintains robust performance (ASR: 0.19%).

For comprehensive evaluation, we also assess both methods on Circuit Breakers' original benchmark scenarios, as shown in Table 8. SafetyLock achieves perfect defense (0% ASR) across all attack types, surpassing Circuit Breakers' performance on its own evaluation metrics.

Table 7: Comparison with Circuit Breakers across different risk levels using Llama-3-8B-Instruct

| Method | Level-1 (ASR/Score) | Level-2 (ASR/Score) | Level-3 (ASR/Score) |
|---|---|---|---|
| Original Fine-tuned | 49.24%/4.13 | 38.46%/3.19 | 42.88%/3.23 |
| Circuit Breakers | 84.62%/3.62 | 27.12%/2.10 | 94.04%/3.79 |
| SafetyLock | **0.19%/1.36** | **5.19%/1.07** | **0.19%/1.04** |

Table 8: Performance on Circuit Breakers' original benchmark scenarios

| Method | AutoDAN | PAIR | GCG |
|---|---|---|---|
| Base Model | 3.7% | 18.7% | 44.5% |
| Circuit Breakers | 0.0% | 7.5% | 2.5% |
| SafetyLock | **0.0%** | **0.0%** | **0.0%** |

Regarding computational efficiency, SafetyLock requires 5 minutes for Meta-SafetyLock construction and 0.1 seconds for distribution to each fine-tuned model. In contrast, Circuit Breakers demands 22 minutes 15 seconds per model on an A100. This significant efficiency advantage, combined with superior safety metrics, demonstrates SafetyLock's practical advantages for large-scale deployment scenarios.

The performance disparity may be attributed to Circuit Breakers' representation remapping strategy being less effective when model safety boundaries have been substantially modified through fine-tuning. SafetyLock's approach of targeting specific attention heads appears more robust to such modifications while maintaining computational efficiency.

### D.7 IMPACT OF TOKEN WINDOW SIZE ON SAFETYLOCK

The choice of how many final tokens to consider when calculating safety directions represents a crucial design decision in SafetyLock's implementation. While previous works often use the entire hidden state for intervention, we hypothesized that focusing on a smaller window of final tokens might capture safety-relevant patterns more effectively while maintaining computational efficiency.

Table 9: Impact of Token Window Size (r) on Safety Performance

| Model | AdvBench ASR (%) | | |
|---|---|---|---|
| | Level-1 | Level-2 | Level-3 |
| Vanilla | 49.24 | 38.46 | 42.88 |
| r = 1 | 1.14 | 6.84 | 3.61 |
| r = 3 | 0.76 | 8.55 | 0.19 |
| r = 5 | **0.19** | **5.19** | **0.19** |
| r = 10 | 0.48 | 8.08 | 0.57 |

To determine the optimal token window size, we conducted extensive experiments varying r from 1 to 10 tokens across all three risk levels, as shown in Table 9. Our findings reveal that r = 5 consistently achieves optimal or near-optimal safety performance across all scenarios. While smaller windows (r = 1, 3) can effectively improve safety, they may not capture sufficient context for robust intervention. Conversely, larger windows (r = 10) show slightly degraded performance, possibly due to including less relevant contextual information. This empirical evidence supports our choice of r = 5 as the default parameter, offering the best balance between robust safety improvement and effective intervention across different fine-tuning scenarios.

# E    RECOMMENDATIONS FOR DEPLOYING SAFETYLOCK

Understanding the diverse landscape of model deployment scenarios is crucial for effectively implementing SafetyLock to maintain safety while enabling customization. The method's effectiveness and implementation strategy vary significantly depending on the model's distribution approach and user priorities, leading to distinct considerations for different deployment contexts.

For closed-source models served through APIs (e.g., GPT-4), SafetyLock offers an optimal solution through seamless integration into the service provider's infrastructure. Model providers can automatically apply SafetyLock after each fine-tuning operation, ensuring consistent safety standards while maintaining customization capabilities. This approach particularly benefits enterprises in regulated industries that require both task-specific optimization and strict safety controls, as it preserves the ability to customize models for specific use cases without compromising safety standards. The automated application of SafetyLock in this context ensures that all model variants maintain robust safety guardrails, regardless of the extent of customization.

In scenarios involving open-source models with safety-conscious users, SafetyLock can be effectively implemented as part of the standard deployment pipeline. Organizations using open-source models can apply SafetyLock during their model serving phase, maintaining safety controls while preserving the benefits of customization. This implementation strategy allows organizations to balance the flexibility inherent in open-source models with the need for robust safety guarantees, ensuring that fine-tuned models remain both useful and safe. Safety-conscious users can leverage SafetyLock to maintain consistent safety standards across their deployments while still benefiting from the customization capabilities that open-source models provide.

To address the fundamental challenge of malicious users with full access to open-source weights, we propose a hybrid deployment strategy that combines transparency with controlled access to safety-critical components. This approach involves open-sourcing the majority of model weights while retaining control of a small subset of safety-critical weights using methods like Taylor Unswift (Wang et al., 2024a). By providing efficient access to these controlled weights through a service API and applying SafetyLock during the serving phase, organizations can maintain crucial safety controls while preserving the benefits of open-source accessibility. This balanced solution ensures that users can customize models for their specific needs without easily circumventing safety measures, as the critical safety-related parameters remain protected under controlled access.

For successful implementation, organizations should establish comprehensive monitoring systems to regularly update safety vectors, implement automatic safety checks post-fine-tuning, and develop clear protocols for handling potential conflicts between safety measures and legitimate use cases. Regular assessment and updating of safety mechanisms ensure that SafetyLock remains effective against evolving harmful behaviors, while clear documentation and guidelines help users understand the implications and importance of these safety measures. Through these carefully considered deployment strategies and best practices, SafetyLock provides a robust framework for maintaining model safety across various deployment scenarios, acknowledging and addressing the inherent challenges in protecting open-source models while enabling their beneficial applications.

