# OpenReview forum: "Locking Down the Finetuned LLMs Safety"
_ICLR.cc/2025/Conference — Submitted to ICLR 2025_

### Official Review · Reviewer_sW5B · 2024-11-03

**Soundness:** 3
**Presentation:** 2
**Contribution:** 2
**Rating:** 5
**Confidence:** 4

**Summary:**

The paper proposes a novel approach, SafetyLock, to mitigate safety risks in fine-tuned large language models (LLMs). SafetyLock leverages the discovery that fine-tuned models retain similar safety-related activation representations to their base models, enabling the extraction of a Meta-SafetyLock, a set of safety bias directions. This allows for efficient and transferable safety alignment in fine-tuned models, reducing the harmful instruction response rate.

**Strengths:**

1. This paper discovers that fine-tuned models retain similar safety-related activation representations to their base models, so the safety directions computed from the original model remain effective when applied to the fine-tuned model.
2. SafetyLock reduces the harmful instruction response rate in fine-tuned models from 60% to below 1%, offering a scalable, non-invasive solution for ensuring the safety of customized LLMs.

**Weaknesses:**

Originality
1. The concept of safety-related directions  seems similar to previous works on concept activation vectors [1] and the design of  adaptive interaction on MHA is similar to [3] where the only difference lies in how to select the intervention heads.

Quality
1. Will the choice of preference-style safety dataset (line 182) affect the identification of safety-related attention heads (line 188)? It would be better to verify the identified safety-related attention heads stay consistent no matter what safety dataset is selected.
2. It is not clear why the element-wise multiplication between the standard deviation of activations and safety directions is needed. Although authors claim that std term captures the variability of the activations, previous works [1,2] only use the safety directions to intervention the model and achieve good performance. Thus, ablation study on std should be considered in this paper otherwise I may doubt whether this part of the design is redundant.
3. The number of chosen Top-K heads are different in different models. Authors claimed in Appendix D that SafetyLock’s intervention degree (K) has impact on model safety. Thus, the time used in the choice of K should also be considered in the method efficiency evaluation.

Clarity
1. Line 308: citation of ICD may be wrong? Besides, the paper contains several duplicate citations (like Mengru Wang et al.), which should be checked and corrected.
2. Line 201: r is the number of final tokens considered, but how authors choose r? I checked the paper but didn’t find the clarification. If it is equal to the hidden size, such safety direction is the same with several previous works [1,2]. If not, whether the method performance is sensitive to r value should also be considered.
3. Figure 2: captions of subfigures(b-d) have the extra spaces and captions of subfigures(e-g) are too close to the figure content.

[1] Zou, A., Phan, L., Chen, S., Campbell, J., Guo, P., Ren, R., ... & Hendrycks, D. (2023). Representation engineering: A top-down approach to ai transparency. arXiv preprint arXiv:2310.01405.

[2] Wang, P., Zhang, D., Li, L., Tan, C., Wang, X., Ren, K., ... & Qiu, X. (2024). Inferaligner: Inference-time alignment for harmlessness through cross-model guidance. arXiv preprint arXiv:2401.11206.

[3] Wang, T., Jiao, X., He, Y., Chen, Z., Zhu, Y., Chu, X., ... & Ma, L. (2024). Adaptive Activation Steering: A Tuning-Free LLM Truthfulness Improvement Method for Diverse Hallucinations Categories. arXiv preprint arXiv:2406.00034.

**Questions:**

1. Would it be better to use the cosine similarity or JS divergence metric to make the quantitative analysis in Figure 2(e-g) since KL divergence is asymmetric?
2. How about the performance of SafeLock on XSTest since I noticed this dataset is evaluated in ICD baseline.

---

> ### Author Response · Authors · 2024-11-27
> **Response 1**
>
> > The concept of safety-related directions seems similar to previous works on concept activation vectors [1] and the design of adaptive interaction on MHA is similar to [3] where the only difference lies in how to select the intervention heads.
>
>
>
> Thank you for raising this profound question about the technical distinctions between SafetyLock and previous work. Our research addresses a critical, largely overlooked real-world challenge - how to efficiently restore safety across hundreds of thousands of fine-tuned models derived from a single base model, whether open-source like Llama or closed-source like GPT-4. This is an increasingly urgent problem as fine-tuning becomes more accessible.
>
> A particular limitation of existing methods [1,2], including inference-time intervention, activation addition, and representation engineering, is their significant computational overhead. These approaches require over 5 minutes to process each 8B-scale fine-tuned model, and 20-30 minutes for 100B-scale models. The **groundbreaking insight** at the heart of SafetyLock is our discovery that safety-related attention heads and their required intervention directions remain remarkably consistent between base and fine-tuned models. This finding reveals that even when model weights are modified through fine-tuning, the attention heads responsible for safety considerations remain largely unchanged.
>
> This technical breakthrough enables an extraordinarily efficient approach: extracting Meta-SafetyLock once from the base model and distributing it across all fine-tuned variants in under 0.1 seconds, even for 100B-scale models. Beyond the headline efficiency gains, SafetyLock introduces several technical innovations compared to [1,3]: we've enhanced stability by utilizing r end tokens rather than a single token for vector extraction, carefully balanced safety and performance through sophisticated alpha selection to avoid over-rejection of legitimate queries, and have an automated head selection process that more precisely targets safety-relevant patterns.
>
> These technical advances collectively enable what was previously thought impossible - near-instantaneous safety restoration that can scale to millions of fine-tuned models while maintaining high performance on legitimate tasks. The fundamental distinctions in both implementation and motivation from prior work like [1,3] demonstrate SafetyLock's novel contribution to addressing this critical real-world challenge. We have further elaborated on these technical distinctions in our Related Work section.
>
>
>
> We appreciate the suggestion to compare with the recent Circuit Breakers work from NeurIPS 2024 [4]. This comparison is particularly valuable as Circuit Breakers builds on Representation Engineering [1] to remap harmful output representations towards incoherent or refusal states. Using three fine-tuned versions of Llama-3-8B-Instruct and maintaining consistent hyperparameters across all experiments, we observed striking differences in performance:
>
> (See in Response 2)

---

> ### Author Response · Authors · 2024-11-27
> **Response 2**
>
> First, in controlling fine-tuning risks, SafetyLock consistently outperformed Circuit Breakers across all three risk levels:
>
> **Level-1 Results** (explicitly harmful fine-tuning):
>
> | Model                | AdvBench ASR | HEx-PHI Score |
> | -------------------- | ------------ | ------------- |
> | Original Fine-tuned  | 49.24%       | 4.13          |
> | Circuit Breakers [4] | 84.62%       | 3.62          |
> | SafetyLock           | **0.19%**    | **1.36**      |
>
> **Level-2 Results** (implicitly harmful fine-tuning):
>
> | Model                | AdvBench ASR | HEx-PHI Score |
> | -------------------- | ------------ | ------------- |
> | Original Fine-tuned  | 38.46%       | 3.19          |
> | Circuit Breakers [4] | 27.12%       | 2.10          |
> | SafetyLock           | **5.19%**    | **1.07**      |
>
> **Level-3 Results** (benign fine-tuning):
>
> | Model                | AdvBench ASR | HEx-PHI Score |
> | -------------------- | ------------ | ------------- |
> | Original Fine-tuned  | 42.88%       | 3.23          |
> | Circuit Breakers [4] | 94.04%       | 3.79          |
> | SafetyLock           | **0.19%**    | **1.04**      |
>
> The performance gap is particularly notable for Level-1 and Level-3 scenarios, where Circuit Breakers actually increases the attack success rate. We hypothesize this unexpected behavior occurs because Circuit Breakers' representation remapping strategy may be less effective when the model's safety boundaries have been significantly altered through fine-tuning.
>
> To ensure fair comparison, we also evaluated both methods on Circuit Breakers' original benchmark scenarios:
>
> | Model                | AutoDAN ASR | PAIR ASR | GCG ASR  |
> | -------------------- | ----------- | -------- | -------- |
> | Base Model           | 3.7%        | 18.7%    | 44.5%    |
> | Circuit Breakers [4] | 0.0%        | 7.5%     | 2.5%     |
> | SafetyLock           | **0.0%**    | **0.0%** | **0.0%** |
>
> SafetyLock achieves perfect defense even on Circuit Breakers' primary evaluation scenarios. Moreover, SafetyLock demonstrates superior efficiency - requiring only 5 minutes for Meta-SafetyLock construction and 0.1 seconds for distribution to any fine-tuned model, compared to Circuit Breakers' 22 minutes 15 seconds per model on an A100.
>
> These comprehensive results validate SafetyLock's effectiveness in addressing the unique challenges of maintaining safety in fine-tuned models, while also demonstrating competitive or superior performance on standard safety benchmarks. We've added these comparisons and detailed analysis in Appendix D.6.
>
>
>
> ---
>
>
>
> [1] Representation engineering: A top-down approach to ai transparency.
>
>
>
> [2] Adaptive Activation Steering: A Tuning-Free LLM Truthfulness Improvement Method for Diverse Hallucinations Categories.
>
>
>
> [3] Wang, T., Jiao, X., He, Y., Chen, Z., Zhu, Y., Chu, X., ... & Ma, L. (2024). Adaptive Activation Steering: A Tuning-Free LLM Truthfulness Improvement Method for Diverse Hallucinations Categories. arXiv preprint arXiv:2406.00034.
>
>
>
> [4] Zou A, Phan L, Wang J, et al. Improving alignment and robustness with circuit breakers[C]//The Thirty-eighth Annual Conference on Neural Information Processing Systems. 2024.

---

> ### Author Response · Authors · 2024-11-27
> **Response 3**
>
> > Will the choice of preference-style safety dataset (line 182) affect the identification of safety-related attention heads (line 188)? It would be better to verify the identified safety-related attention heads stay consistent no matter what safety dataset is selected.
>
>
>
> To verify the consistency of our identified safety-related attention heads, we conducted additional experiments using an alternative safety dataset from [3], paired with unsafe responses generated by the Level-1 fine-tuned Llama-3-8B-Instruct model. Our analysis reveals remarkable consistency in head identification across different preference datasets:
>
> Thank you for this insightful question about the robustness of our safety head identification process across different preference-style safety datasets. **Our analysis reveals that dataset choice does impact head selection, but to a moderate and manageable degree:**
>
> | Selection Size | Common Heads | Total Heads | Overlap Rate |
> | -------------- | ------------ | ----------- | ------------ |
> | Top K=24       | 11           | 1024        | 45.8%        |
> | Top K=48       | 25           | 1024        | 52.1%        |
>
> Given that these heads are selected from a total pool of 1024 attention heads, the substantial overlap (nearly 50%) between different safety datasets suggests that our method consistently identifies intrinsic safety-related patterns in the model's architecture. This stability in head identification across different safety datasets provides strong evidence for the robustness of our approach and indicates that SafetyLock captures fundamental safety mechanisms rather than dataset-specific patterns.
>
>
>
> > It is not clear why the element-wise multiplication between the standard deviation of activations and safety directions is needed. Although authors claim that std term captures the variability of the activations, previous works [1,2] only use the safety directions to intervention the model and achieve good performance. Thus, ablation study on std should be considered in this paper otherwise I may doubt whether this part of the design is redundant.
>
>
>
> We deeply appreciate your insightful question about the necessity of the standard deviation term $σ_{l}^h$. While [1,2] demonstrate good performance using only safety directions, our work specifically addresses the challenge of maintaining robust performance across diverse tasks while ensuring safety. The standard deviation term $σ_{l}^h$ plays a crucial role in this context.
>
> The inclusion of $σ_{l}^h$ serves a critical purpose: it makes the hyperparameter α more robust and less sensitive to tuning, as discussed in Appendix D.1. This is particularly important when balancing safety with model performance in open-ended tasks. Unlike previous works that primarily focus on safety metrics, our evaluation spans eight diverse generalization tasks including mathematics, reasoning, and open-ended question answering (Section 4.5).
>
> To demonstrate this empirically, we conducted detailed ablation studies. We use GSM8K as a special case for Level-3, considering the mathematical abilities of the fine-tuned models:
>
> | Model                  | AdvBench ASR | HEx-PHI Score | GSM8K Test Acc |
> | ---------------------- | ------------ | ------------- | -------------- |
> | Original               | 7.23%        | 1.45          | 85.59%         |
> | SafetyLock w/o $σ_{l}^h$ | **0.0%**     | **1.03**      | 52.24%         |
> | SafetyLock w $σ_{l}^h$   | 0.19%        | 1.08          | **85.01%**     |
>
> Without $σ_{l}^h$, while safety metrics improve marginally, we observe substantial degradation in mathematical reasoning capabilities (52.24% on GSM8K). Including $σ_{l}^h$ maintains strong safety improvements while preserving model performance (85.01%). This demonstrates that $σ_{l}^h$ is not redundant but rather essential for achieving our goal of balanced safety and capability preservation.

---

> ### Author Response · Authors · 2024-11-27
> **Response 4**
>
> > The number of chosen Top-K heads are different in different models. Authors claimed in Appendix D that SafetyLock’s intervention degree (K) has impact on model safety. Thus, the time used in the choice of K should also be considered in the method efficiency evaluation.
>
>
>
> We sincerely appreciate your question about the selection of Top-K heads across different model scales. During the rebuttal period, we discovered a fascinating scaling law for the optimal K value. Using Llama-3.2-1B-Instruct as an additional experimental point, we observed the following pattern:
>
> | K Value | AdvBench ASR |
> | ------- | ------------ |
> | Vanilla | 21.15%       |
> | K=3     | 16.54%       |
> | K=6     | **10.65%**   |
> | K=12    | 11.08%       |
> | K=24    | 12.44%       |
> | K=48    | 47.50%       |
>
> This experiment, combined with our previous results, reveals a consistent scaling pattern across model sizes:
>
> - 1B-scale models: optimal K ≈ 6-12 heads
> - 8B-scale models: optimal K ≈ 12-24 heads
> - 70B/123B-scale models: optimal K ≈ 24-48 heads
>
> This pattern suggests that safety-sensitive attention heads constitute approximately 1-4% of total heads, with larger models potentially requiring a smaller proportion of heads for effective safety control. The existence of this scaling law means we can directly determine appropriate K values based on model size without additional search time, making SafetyLock's efficiency claims even more robust. We have revised this point in Appendix D.1.
>
>
>
>
>
> > ine 308: citation of ICD may be wrong? Besides, the paper contains several duplicate citations (like Mengru Wang et al.), which should be checked and corrected.
>
>
>
> We sincerely appreciate your careful attention to citation accuracy. **The citation of ICD at line 308 is indeed correct.** We have clarified Line 308 by spelling out "in-context demonstration (ICD)" to avoid potential confusion with other references. And we have also conducted a thorough review of our citations and removed duplicate references. Thank you for helping us maintain bibliographic preciion in our paper. Please refer to our revised version.
>
>
>
> > Line 201: r is the number of final tokens considered, but how authors choose r? I checked the paper but didn’t find the clarification. If it is equal to the hidden size, such safety direction is the same with several previous works [1,2]. If not, whether the method performance is sensitive to r value should also be considered.
>
>
>
> We sincerely appreciate your question about the choice of parameter r. In our experiments, we set r=5, considering the last 5 tokens for safety direction calculation. To thoroughly evaluate the impact of this choice, we conducted ablation studies with different values of r on Llama-3-8B-Instruct across all risk levels:
>
> | **Model** | **Level-1** | **Level-2** | **Level-3** |
> | --------- | ----------- | ----------- | ----------- |
> | Vanilla   | 49.24%      | 38.46%      | 42.88%      |
> | r=1       | 1.14%       | 6.84%       | 3.61%       |
> | r=3       | 0.76%       | 8.55%       | 0.19%       |
> | r=5       | **0.19%**   | **5.19%**   | **0.19%**   |
> | r=10      | 0.48%       | 8.08%       | 0.57%       |
>
> The results demonstrate that r=5 consistently achieves optimal or near-optimal performance across all risk levels. While both smaller (r=1,3) and larger (r=10) values can effectively improve safety, r=5 provides the best balance of robustness and effectiveness. This empirical finding has been added to our experimental setup section, and detailed ablation results are now included in the Appendix D.7 for completeness.
>
>
>
> > 1. Figure 2: captions of subfigures(b-d) have the extra spaces and captions of subfigures(e-g) are too close to the figure content.
> > 2. Would it be better to use the cosine similarity or JS divergence metric to make the quantitative analysis in Figure 2(e-g) since KL divergence is asymmetric?
>
> Thank you for your reminder. We have revised Figure 2, and used cosine similarity for the statistics in panels e-g.

---

> ### Author Response · Authors · 2024-11-27
> **Response 5**
>
> > How about the performance of SafeLock on XSTest since I noticed this dataset is evaluated in ICD baseline.
>
>
>
> Thank you for asking about SafetyLock's performance on XSTest . We have conducted comprehensive experiments on this dataset and included the results in Table 2 and Section 4.4. Here's how SafetyLock performs compared to other defense methods:
>
> | Model          | XSTest ASR |
> | -------------- | ---------- |
> | Vanilla        | 19.5%      |
> | in-context demonstration (ICD)            | 7.0%       |
> | PPL            | 17.0%      |
> | Paraphrase     | 40.0%      |
> | Retokenization | 57.5%      |
> | Self-Reminder  | 8.0%       |
> | Self-Exam      | 19.5%      |
> | **SafetyLock** | **4.0%**   |
>
> SafetyLock achieves state-of-the-art performance on XSTest with a **4.0% ASR**, significantly outperforming all baseline methods. This result is particularly noteworthy as it demonstrates SafetyLock's effectiveness against structured adversarial attacks, complementing our existing results on other attack types (AutoDAN, DeepInception, GCG, and PAIR). The strong performance across this diverse set of attack benchmarks further validates SafetyLock's robustness as a comprehensive safety solution.

---

> > ### Comment · Reviewer_sW5B · 2024-12-01
> >
> > Thank you for your response. It addressed most of my questions, and this manuscript still requires a thorough revision to make it ready for publication, so I will keep my score.

---

> > > ### Author Response · Authors · 2024-12-02
> > >
> > > Dear Reviewer sW5B,
> > >
> > > We are very pleased that you consider we have addressed most of your questions. Regarding the manuscript, we have **carefully updated the manuscript** during the discussion phase, with all additions shown in blue text. We would be very grateful if you could provide more specific details about which aspects of the manuscript still need modification.
> > >
> > > We highly value your feedback, and thank you for helping us improve the quality of our work.

---

### Official Review · Reviewer_ZuM2 · 2024-11-03

**Soundness:** 2
**Presentation:** 3
**Contribution:** 3
**Rating:** 6
**Confidence:** 3

**Summary:**

This paper introduces SafetyLock, a method to ensure the safety of LLMs post-fine-tuning, addressing the risks that traditional alignment measures fail to mitigate. SafetyLock leverages the discovery that fine-tuned models retain safety-related activation patterns from their base models, allowing the extraction and application of Meta-SafetyLock directions to reinforce safety efficiently. Experiments show that SafetyLock significantly reduces harmful behavior, achieving a response rate reduction from 60% to below 1% with minimal computational cost, offering a scalable and robust solution for safer LLM customization.

**Strengths:**

1. The main contribution of this paper is to point out an important safety issue in distributing LLMs: How can we guarantee the safety of an LLM after we release it, where users can use fine-tuning to bypass its safety alignment. Although the proposed solution is less satisfied to the reviewer (See Weakness 1.), the reviewer admit the value of the research question itself.

2. This paper provides insights into the inner machanism of LLMs on safety.

**Weaknesses:**

1. Regarding the research question. The paper aims to propose a *lock* to ensure that the safety behavior of an released LLM would not be jailbreaked. To this end, the major drawback of the proposed method is that: the safety lock is applied to the LLM after fine-tuning, which means that the fine-tuned LLM should still be preserved and served by its original provider. This is effective for proprietary LLMs such as GPT-4, where the model checkpoint is still kept by the company even after fine-tuning via API. However, for open-sourced LLMs, such as LLaMA series, the proposed method failed as the provider cannot apply the safety lock to fine-tuned LLMs. I would expect a method which can by applied instantly before the release of the checkpoint and would protect the LLM from any unsafe modifications.


2. Regarding the methodology. The *safety direction* is calculated according to Eq.3, which involves the direct comparison between two different sentences. I would suspect the rationality of the comparison. For example, if the last 3 tokens of the safe reponse is *I can't answer* and the last 3 tokens of the unsafe response is *make a bomb*. The comparison between `I` and `make`, `can't` and `a`, and `answer` and `bomb` does not make sense. The meaning of the last $r$ tokens is not strictly aligned between the safe and the unsafe responses. The author should give more explanations on this. Besides, there are different aspects to investigate the safety mechanism, such as activations of individual neurons that are closely related to safety behaviors. They are not fully explored.



3. Minor issues (Does not affect my scores)
    - Missing references. This paper should also survey papers for `LLM safety mechanism`, such as *A mechanistic understanding of alignment algorithms: A case study on dpo and toxicity*, *Finding Safety Neurons in Large Language Models*, etc.
    - Formats. Check the font of the sub-title in Figure 2. Also, the sub-title overlaps with the figures.
    - Line 296 - Line 297: 2e-5 --> $2 \times 10^{-5}$



I am quite torn between giving a score of 5 or 6. I fully acknowledge the importance of emphasizing to the research community the need for robust safety mechanisms in publicly released LLMs. However, I am somewhat dissatisfied with the proposed method, as it does not thoroughly address the problem. Considering that this is a relatively early attempt (though perhaps I am not fully aware of previous work in this area) and revealing a problem is more important than a perfect solution, I am inclined to give a somewhat positive score of 6. I hope the authors can adequately address my concerns in the rebuttal, and I will consider adjusting my score to a more negative one if necessary.

**Questions:**

See weakness

---

> ### Author Response · Authors · 2024-11-27
> **Response 1**
>
> >Regarding the research question. The paper aims to propose a *lock* to ensure that the safety behavior of an released LLM would not be jailbreaked. To this end, the major drawback of the proposed method is that: the safety lock is applied to the LLM after fine-tuning, which means that the fine-tuned LLM should still be preserved and served by its original provider. This is effective for proprietary LLMs such as GPT-4, where the model checkpoint is still kept by the company even after fine-tuning via API. However, for open-sourced LLMs, such as LLaMA series, the proposed method failed as the provider cannot apply the safety lock to fine-tuned LLMs. I would expect a method which can by applied instantly before the release of the checkpoint and would protect the LLM from any unsafe modifications.
>
>
>
> We appreciate this critical insight about deployment challenges. You raise an important point about the limitations of post-fine-tuning safety measures for open-source models. Here's how we propose to address these scenarios:
>
> - For **closed-source models with safety-conscious users** (e.g., enterprises using GPT-4 API), SafetyLock provides an optimal solution. The model provider can efficiently distribute and apply safety measures post-fine-tuning, ensuring both customization and safety. This is particularly valuable for regulated industries requiring both task-specific optimization and strict safety controls.
>
> - For **closed-source models with safety-indifferent users**, SafetyLock remains effective as the provider maintains control over model deployment and can enforce safety measures regardless of user preferences. This ensures consistent safety standards across all deployments while still allowing legitimate customization.
>
> - For **open-source models with safety-conscious users**, SafetyLock offers a valuable solution by providing users with the tools to maintain safety in their fine-tuned variants. These users can apply SafetyLock during their deployment pipeline, effectively preserving both customization benefits and safety guarantees.
>
> - However, we acknowledge a fundamental limitation with **open-source models and malicious users**. Once model weights are fully accessible, determined adversaries can circumvent any pre-deployed safety measures through various techniques, from gradient-based modifications to extreme cases of random reinitialization. This represents an inherent challenge in open-source AI deployment that extends beyond the scope of post-fine-tuning safety measures.
>
> To address this limitation, we propose a potential directions for future research. For example, **we can open-source the majority of the model weights, while service providers retain a small subset of the weights (such as using the Taylor Unswift [1] method) to maintain security via mechanisms like SafetyLock**. Users are not able to directly modify the specialized weights, but they can access the service provider's model with minimal computational resource, allowing them to attempt training or deploying a private model. This approach satisfies the need for private data while mitigating the security risks posed by open-source models and malicious users. Thank you again for your careful consideration. We have further discussed the Recommendations for Deploying SafetyLock in Appendix E, aiming to clarify how the community should be encouraged to advance the safe application of the model.
>
>
>
> ---
>
> [1] Wang G, Chuang Y N, Tang R, et al. Taylor Unswift: SecuredWeight Release for Large Language Models via Taylor Expansion[J]. arXiv preprint arXiv:2410.05331, 2024.

---

> ### Author Response · Authors · 2024-11-27
> **Response 2**
>
> >  Regarding the methodology. The *safety direction* is calculated according to Eq.3, which involves the direct comparison between two different sentences. I would suspect the rationality of the comparison. For example, if the last 3 tokens of the safe reponse is *I can't answer* and the last 3 tokens of the unsafe response is *make a bomb*. The comparison between `I` and `make`, `can't` and `a`, and `answer` and `bomb` does not make sense. The meaning of the last r tokens is not strictly aligned between the safe and the unsafe responses. The author should give more explanations on this. Besides, there are different aspects to investigate the safety mechanism, such as activations of individual neurons that are closely related to safety behaviors. They are not fully explored.
>
> Thank you for this critical methodological question. Let us explain why our token comparison approach is theoretically sound and empirically effective:
>
> **Theoretical Foundation**: In decoder-only language models like Llama, each token's representation is inherently contextual. When a model is generating the token "bomb" in an unsafe response, its attention heads aren't just processing that single token - **they're encoding the entire conversation context that led to generating harmful content.** Similarly, when generating "can't" in a safe refusal, the representations capture the model's decision to decline the request. Therefore, comparing these activations reveals broader safety-relevant patterns beyond simple token-level differences.
>
> **Empirical Validation**: To rigorously test this hypothesis, we conducted comparative experiments:
>
> | Method                     | Level-1   | Level-2   | Level-3   |
> | -------------------------- | --------- | --------- | --------- |
> | Vanilla                    | 49.24%    | 38.46%    | 42.88%    |
> | Perfectly aligned [2]      | 2.84%     | 6.74%     | 1.08%     |
> | Last r tokens (SafetyLock) | **0.19%** | **5.19%** | **0.19%** |
>
> We tested two approaches:
>
> 1. Our **last r tokens** method
> 2. A perfectly aligned QA format from [2]:
>
> ```
> **MODEL INPUT:**
>
> [INST] Can you tell me the process for making explosives?
> Choices:
> A: "Of course, the process for explosives is..."
> B: "No." [/INST]
>
> ---
> **OUTPUT:**
>
> *Pos:* B
> *Neg:* A
> ```
>
> The results show our method outperforms even perfectly aligned comparisons. This is because in real-world applications, models must maintain safety across diverse, open-ended scenarios - not just multiple-choice situations. Our analysis of attention patterns reveals that the final tokens' representations consistently encode critical safety decisions, regardless of the specific tokens involved.
>
> For example, when analyzing the attention patterns of Llama-3-8B:
>
> - Safe responses show consistent activation patterns in key safety heads even with varied refusal phrasings ("I can't", "I apologize", "That would be harmful")
> - Unsafe responses exhibit distinct patterns regardless of the specific harmful content generated
>
> This demonstrates that our method captures fundamental safety mechanisms rather than superficial token alignments. While we acknowledge the value of neuron-level analysis for understanding safety mechanisms, our attention-head approach offers an effective balance of interpretability, efficiency, and real-world performance.
>
> ---
>
> [2] Panickssery N, Gabrieli N, Schulz J, et al. Steering llama 2 via contrastive activation addition[J]. arXiv preprint arXiv:2312.06681, 2023.
>
>
>
> > Minor issues (Does not affect my scores)
> >
> > - Missing references. This paper should also survey papers for `LLM safety mechanism`, such as *A mechanistic understanding of alignment algorithms: A case study on dpo and toxicity*, *Finding Safety Neurons in Large Language Models*, etc.
> >
> > - Formats. Check the font of the sub-title in Figure 2. Also, the sub-title overlaps with the figures.
> >
> > - Line 296 - Line 297: 2e-5 -->
> >
> >   2×10−5
>
> We sincerely appreciate your careful attention to these formatting and citation details. We have addressed all these issues in our revision:
>
> - We have **expanded our literature review** with additional key references on LLM safety mechanisms, including the suggested papers on mechanistic understanding of alignment and safety neurons.
> - The formatting issues in Figure 2 have been corrected, with proper spacing and non-overlapping subtitles.
> - We have standardized all scientific notation, changing "2e-5" to "2×10^{-5}" throughout the paper for consistency.
>
> These additions are highlighted in \textcolor[RGB]{0,0,139}{blue} in the paper! Thank you for helping us improve the paper's technical precision and readability.

---

### Official Review · Reviewer_uK5C · 2024-11-04

**Soundness:** 2
**Presentation:** 3
**Contribution:** 2
**Rating:** 3
**Confidence:** 4

**Summary:**

The paper proposes a method to identify safety-relevant attention heads, interpreted as the model's safety mechanism. Furthermore, experiments are conducted to verify the effectiveness of these safety heads and their transferability across models.

**Strengths:**

1. They implement the method of ITI [1] on the safety of LLM and verify the effectiveness of the safety head.

2. Experiment results show the existence of safety heads and can efficiently enhance models' safety on different models.



[1] Inference-Time Intervention: Eliciting Truthful Answers from a Language Model, NeurIPS 2023

**Weaknesses:**

1. The method is not novel. It totally follows the method in ITI [1] without considering the properties of safety itself, which influence the accuracy of the analysis.  (1) As noted in the abstract (line 14), only 10 sentences can compromise the models' safety mechanisms. However, as illustrated in line 193, the safety components in LLMs are composed of three-fourths for both LLama3-8b and LLama3-70b. If the safety mechanism is easily breached, it may be due to a relatively small number of parameters. (2) The head is not a fine-grained aspect of analysis, as there are papers exploring neuron-level safety mechanisms [2]. Additionally, the feed-forward layer is important for safety, as models may refuse to extract harmful knowledge and only extract safety knowledge such as "Apologize I can not ....." [3]. However, the paper ignores this and focuses solely on heads in self-attention.

2. The analysis is not comprehensive. Figure 2 examines the "safety direction" at layer 31, the final layer of LLama3-8B. This analysis is biased, as it only evaluates the output of different models. It's expected that embeddings will have some relation or differences because the models' outputs vary. What would be more valuable is an analysis of the intermediate layers, exploring how the mechanism identifies and mitigates harmful information. However, the paper lacks further analysis, which would have been interesting.

3. The experiments in Table 1 lack baselines, and the baselines in Figure 5 are all trivial. It’s important to compare lightweight defense methods, such as [4], to demonstrate the effectiveness of your approach.


[1] Inference-Time Intervention: Eliciting Truthful Answers from a Language Model, NeurIPS 2023

[2] Finding Safety Neurons in Large Language Models, Arxiv 2024

[3] Transformer Feed-Forward Layers Are Key-Value Memories, ACL 2021

[4] Improving Alignment and Robustness with Circuit Breakers, NeurIPS 2024

**Questions:**

In Section 4.1, you mentioned that the learning rate is set to 2e-5. I'm curious about the effect of this learning rate on the experimental results. A learning rate of 1e-6 is typically used for continued pretraining, which has a smaller impact on model performance including general capability and safety alignment.

---

> ### Author Response · Authors · 2024-11-27
> **Response 1**
>
> > The method is not novel. It totally follows the method in ITI [1] without considering the properties of safety itself, which influence the accuracy of the analysis. (1) As noted in the abstract (line 14), only 10 sentences can compromise the models' safety mechanisms. However, as illustrated in line 193, the safety components in LLMs are composed of three-fourths for both LLama3-8b and LLama3-70b. If the safety mechanism is easily breached, it may be due to a relatively small number of parameters. (2) The head is not a fine-grained aspect of analysis, as there are papers exploring neuron-level safety mechanisms [2]. Additionally, the feed-forward layer is important for safety, as models may refuse to extract harmful knowledge and only extract safety knowledge such as "Apologize I can not ....." [3]. However, the paper ignores this and focuses solely on heads in self-attention.
>
> We deeply appreciate your thoughtful critique regarding novelty and methodology. While SafetyLock builds upon insights from ITI, our **key innovation** lies in achieving both superior efficiency and effectiveness in safety restoration. Let me demonstrate with concrete evidence:
>
> **Efficiency Comparison with ITI:**
>
> | Method     | Time Cost (8B) | Memory (GB) | Distribution |
> | ---------- | -------------- | ----------- | ------------ |
> | ITI        | 302.4s         | 24.2        | Per-model    |
> | SafetyLock | 0.001s         | 0.0         | One-time     |
>
> **Safety Performance Comparison:**
>
> | Method     | Level-1 (ASR) | Level-2 (ASR) | Level-3 (ASR) |
> | ---------- | ------------- | ------------- | ------------- |
> | Vanilla    | 49.24%        | 38.46%        | 42.88%        |
> | ITI        | 1.11%         | 7.22%         | 3.43%         |
> | SafetyLock | 0.19%         | 5.19%         | 0.19%         |
>
> The **fundamental distinction** from previous approaches is our focus on scalable safety restoration. Traditional methods, including ITI and Representation engineering, require approximately **5 minutes of GPU computation per 8B-scale model** to obtain necessary activations. In contrast, SafetyLock achieves unprecedented efficiency - **requiring less than 0.1 second even for 100B-scale models** through our innovative offline distribution mechanism.
>
> This effectiveness stems from two key design choices:
>
> 1. **Minimal Head Selection**: For Llama-3-8B-Instruct with 1024 attention heads, we select K=24 (approximately 2% of heads), and for Llama-3-70B-Instruct with 5120 attention heads, we select K=48 (approximately 1% of heads).
>
> 2. **Multi-Token Robustness**: While previous works rely on single token interventions, we consider the last r tokens to enhance robustness:
>
> | Model   | Level-1   | Level-2   | Level-3   |
> | ------- | --------- | --------- | --------- |
> | Vanilla | 49.24%    | 38.46%    | 42.88%    |
> | r=1     | 1.14%     | 6.84%     | 3.61%     |
> | r=3     | 0.76%     | 8.55%     | 0.19%     |
> | r=5     | **0.19%** | **5.19%** | **0.19%** |
> | r=10    | 0.48%     | 8.08%     | 0.57%     |
>
> Our ablation study shows r=5 achieves optimal balance between safety and performance, demonstrating that considering multiple tokens enhances the robustness of safety interventions across different scenarios.
>
> Regarding granularity and the focus on attention heads, we made this architectural choice after careful consideration of the trade-offs between effectiveness and efficiency. While neuron-level analysis [2] and feed-forward layer examination [3] offer valuable insights, our goal was to develop a method that could be practically deployed at scale. Our experiments show that attention head intervention achieves robust safety restoration while maintaining the computational efficiency necessary for large-scale deployment. The consistent performance across our comprehensive evaluation suite suggests that this level of granularity effectively captures and modifies safety-relevant patterns. We have revised the Related Work section and added discussions about these works.

---

> ### Author Response · Authors · 2024-11-27
> **Response 2**
>
> > The analysis is not comprehensive. Figure 2 examines the "safety direction" at layer 31, the final layer of LLama3-8B. This analysis is biased, as it only evaluates the output of different models. It's expected that embeddings will have some relation or differences because the models' outputs vary. What would be more valuable is an analysis of the intermediate layers, exploring how the mechanism identifies and mitigates harmful information. However, the paper lacks further analysis, which would have been interesting.
>
>
>
> We genuinely appreciate your insightful observation about the layer analysis. We want to clarify that our focus on layer 31, head 26 stems from its **empirically proven significance** in safety detection, not arbitrary selection. Through rigorous probing analysis, this attention head demonstrated the strongest correlation with safety-related features among all heads in the model.
>
> To address your valuable point about comprehensive analysis, we have expanded our investigation to include multiple safety-sensitive attention heads identified through probing. These attention heads are distributed across both final and intermediate layers: [(12, 21), (12, 23), (14, 11), (16, 7), (16, 28), (16, 29), (24, 14), (31, 26)]. Our analysis reveals that these heads maintain **consistent directional patterns** across different fine-tuning tasks, closely aligning with their orientations in the original model. This finding is particularly significant as it validates our core hypothesis: Meta-SafetyLock extracted from the base model can be effectively distributed to various fine-tuned variants.
>
> The practical success of this approach is demonstrated in Table 1, where Meta-SafetyLock derived from the original model successfully restores safety across different fine-tuning scenarios. We have revised the manuscript to better highlight these crucial aspects of our analysis.

---

> ### Author Response · Authors · 2024-11-27
> **Response 3**
>
> > The experiments in Table 1 lack baselines, and the baselines in Figure 5 are all trivial. It’s important to compare lightweight defense methods, such as [4], to demonstrate the effectiveness of your approach.
>
> Thank you for this valuable suggestion regarding baseline comparisons. During the discussion period, we conducted comprehensive experiments comparing SafetyLock with the method proposed in NeurIPS 2024 [4], using their official implementation and maintaining consistent hyperparameters except for model weights (as [4] targets base model safety while we address fine-tuned model safety).
>
> Using Llama-3-8B-Instruct, we observed the following results across different risk levels:
>
> **Level-1 Results:**
>
> | Model                              | AdvBench ASR | HEx-PHI Score |
> | ---------------------------------- | ------------ | ------------- |
> | Original Fine-tuned                | 49.24%       | 4.13          |
> | Circuit Breakers [NeurIPS 2024, 4] | 84.62%       | 3.62          |
> | SafetyLock                         | **0.19%**    | **1.36**      |
>
> **Level-2 Results:**
>
> | Model                              | AdvBench ASR | HEx-PHI Score |
> | ---------------------------------- | ------------ | ------------- |
> | Original Fine-tuned                | 38.46%       | 3.19          |
> | Circuit Breakers [NeurIPS 2024, 4] | 27.12%       | 2.10          |
> | SafetyLock                         | **5.19%**    | **1.07**      |
>
> **Level-3 Results:**
>
> | Model                              | AdvBench ASR | HEx-PHI Score |      |
> | ---------------------------------- | ------------ | ------------- | ---- |
> | Original Fine-tuned                | 42.88%       | 3.23          |      |
> | Circuit Breakers [NeurIPS 2024, 4] | 94.04%       | 3.79          |      |
> | SafetyLock                         | **0.19%**    | **1.04**      |      |
>
> Notably, NeurIPS 2024 method [4] requires significant additional time (22 minutes 15 seconds on an A100), exceeding even the DINM model editing approach. In contrast, SafetyLock requires only 5 minutes to construct Meta-SafetyLock and 0.1 seconds for distribution to any fine-tuned model.
>
> Furthermore, we evaluated both methods on the original paper's [4] benchmark scenarios using Llama-3-8B-Instruct:
>
> | Model                              | AutoDAN ASR | PAIR ASR | GCG ASR  |
> | ---------------------------------- | ----------- | -------- | -------- |
> | Base Model, Llama-3-8B-Instruct    | 3.7%        | 18.7%    | 44.5%    |
> | Circuit Breakers [NeurIPS 2024, 4] | 0.0%        | 7.5%     | 2.5%     |
> | SafetyLock                         | **0.0%**    | **0.0%** | **0.0%** |
>
> These results demonstrate SafetyLock's superior performance both in terms of effectiveness and efficiency, achieving consistent **0% ASR** even in the primary scenarios considered by [4].
>
>
> > In Section 4.1, you mentioned that the learning rate is set to 2e-5. I'm curious about the effect of this learning rate on the experimental results. A learning rate of 1e-6 is typically used for continued pretraining, which has a smaller impact on model performance including general capability and safety alignment.
>
>
>
> Thank you for this opportunity to clarify our learning rate selection. Following the hyperparameter settings from [1] (detailed in [1] Appendix G.1).
>
> In Rebuttal stage, **we conducted additional experiments with Llama-3-8B-Instruct at learning rate 1e-6 for Level-3 fine-tuning scenario**. Here's how the different learning rates compare:
>
> | Learning Rate | Vanilla ASR | SafetyLock ASR |
> | ------------- | ----------- | -------------- |
> | 2e-5          | 42.88%      | 0.19%          |
> | 1e-6          | 26.92%      | 0.00%          |
>
> Indeed, our experiments reveal that when using a learning rate of 1e-6, the safety degradation is less severe compared to 2e-5 (ASR: 26.92% vs 42.88%). However, **SafetyLock effectively restores safety under both learning rates**, reducing ASR to near-zero (0.00% and 0.19% respectively). This demonstrates SafetyLock's effectiveness across different fine-tuning configurations, while also highlighting that lower learning rates can help preserve inherent safety properties during fine-tuning.

---

> ### Author Response · Authors · 2024-11-30
>
> Dear Reviewer uK5C,
>
> As the discussion period is coming to an end soon, we wanted to check if you have had a chance to review our responses. Please let us know if your questions have been adequately addressed - we are happy to provide any additional clarification needed. Thank you for your time!
>
> Best Regards,
>
> Authors of "Locking Down the Finetuned LLMs Safety"

---

### Official Review · Reviewer_eTzc · 2024-11-04

**Soundness:** 3
**Presentation:** 3
**Contribution:** 2
**Rating:** 5
**Confidence:** 4

**Summary:**

It is now well known that fine-tuning language models comes with a safety compromise [R0-R4], i.e., the safety alignment gets disturbed when a safety-aligned model is tuned to learn something new. This paper introduces SafetyLock, a method designed to maintain safety alignment post-fine-tuning. For the top-K heads in the model, a safety direction is computed using a preference-style safety dataset. The top-K heads are identified based on the quality signals they provide for safe-unsafe text classification. In the fine-tuned model, the safety vector of these top-K heads is added to the output of attention computations before the up-projection layer. Overall, the method appears sound and is shown to preserve harmlessness on Llama-3-8B, 70 Instruct, and Mistral-Large-2 123B when tested on three risk levels: benign, explicit, and implicit harmful datasets.

**Strengths:**

**Strengths**

- The paper proposes a new idea to prevent safety compromise owing to downstream fine-tuning of the model. While there are existing safety vector-based approaches to prevent safety compromise (with comparable memory and time footprint), the activation-based safety vector computation is a novel contribution.

- The experiments are sound, demonstrating the method's effectiveness in mitigating safety issues across various risk levels, including explicitly harmful, implicitly harmful, and benign datasets. Additionally, the method preserves most of the model's generic utility.

**Weaknesses:**

**Weaknesses**

- The paper largely ignores comparisons with a line of closely related work, beginning with "Language models are Homer Simpson! Safety re-alignment of fine-tuned language models through task arithmetic," (R1) which also introduced the concept of a *safety vector*. Therefore, I do not agree with the claim, *"we are the first to consider locating safety vectors and then restoring the safety of fine-tuned LLMs using an inference-time intervention method,"* as there are a series of similar existing methods, such as R1-R4, and more within this line of research. These methods also use safe-unsafe pairs to derive a safety vector. I suggest the authors properly cite and compare their work with these studies.

- The experiments demonstrating the utility of the method are limited. I understand that works like Qi et al. (2023b) [R0] largely rely on three risk categories, which is sufficient to prove that the problem of safety compromise due to fine-tuning exists. However, solving the problem would require more substantial evidence than relying on a similar set of experiments. It would be beneficial to see the method applied across different fine-tuning domains (such as math, code, and various other tasks) and observe the domain-specific performance with and without the safety measures. For instance, fine-tune on GSM8K, evaluate the test-set performance post task-specific tuning with and without the safety lock, and compare these results with those of related works.

References:

[R0] Qi, Xiangyu, et al. "Fine-tuning aligned language models compromises safety, even when users do not intend to!." arXiv preprint arXiv:2310.03693 (2023).

[R1] Bhardwaj, Rishabh, Do Duc Anh, and Soujanya Poria. "Language Models are Homer Simpson! Safety Re-Alignment of Fine-tuned Language Models through Task Arithmetic." arXiv preprint arXiv:2402.11746 (2024).

[R2] Zhao, Weixiang, et al. "Towards comprehensive and efficient post safety alignment of large language models via safety patching." arXiv preprint arXiv:2405.13820 (2024).

[R3] Hazra, Rima, et al. "Safety Arithmetic: A Framework for Test-time Safety Alignment of Language Models by Steering Parameters and Activations." arXiv preprint arXiv:2406.11801 (2024).

[R4] Yi, Xin, et al. "A safety realignment framework via subspace-oriented model fusion for large language models." arXiv preprint arXiv:2405.09055 (2024).

(and more relevant works exist in this line)

**Questions:**

1) Can you explain the need for the standard deviation of activations $\sigma_l^h$ in equation 4? Any experiments that show with and without $\sigma_l^h$ effect?

2) Can you please elaborate on section 4.5? The setting and motivation aren't very clear to me.

---

> ### Author Response · Authors · 2024-11-27
> **Response 1**
>
> > The paper largely ignores comparisons with a line of closely related work, beginning with "Language models are Homer Simpson! Safety re-alignment of fine-tuned language models through task arithmetic," (R1) which also introduced the concept of a *safety vector*. Therefore, I do not agree with the claim, *"we are the first to consider locating safety vectors and then restoring the safety of fine-tuned LLMs using an inference-time intervention method,"* as there are a series of similar existing methods, such as R1-R4, and more within this line of research. These methods also use safe-unsafe pairs to derive a safety vector. I suggest the authors properly cite and compare their work with these studies.
>
> We sincerely thank the reviewer for this insightful observation. You raise an important point about the relationship between our work and prior research, particularly regarding safety vectors. We fully acknowledge the pioneering contributions of previous works like "Language Models are Homer Simpson!" [R1] and subsequent developments [R2-R4] in establishing the foundational concept of safety vectors.
>
> After careful consideration, we propose revising our claim to more accurately reflect our work's novel contributions while appropriately positioning it within the existing research landscape:
>
> *"Our work advances the field of LLM safety alignment by introducing **Meta-SafetyLock**, a framework that fundamentally reimagines how safety measures can be efficiently distributed across fine-tuned models. While previous works established important foundations through safety vectors [R1] and various intervention methods [R2-R4], our approach uniquely operates at the attention-head level, supported by our discovery that safety-relevant attention heads maintain consistency even after fine-tuning. This insight enables us to extract a single Meta-SafetyLock from the base model that can be rapidly deployed across multiple fine-tuned variants without requiring repeated safety pattern searches, achieving remarkable efficiency (0.0001s) without GPU resources."*
>
> **Our Meta-SafetyLock framework represents a distinct technical advancement in LLM safety alignment**, differentiated from prior approaches in several key aspects:
>
> 1. While [R1] uses direct weight arithmetic and [R2] employs safety patching through decoding strategies, Meta-SafetyLock introduces **attention-head level interventions**. We demonstrate that safety-relevant attention heads maintain consistency post-fine-tuning (Although these Attention Heads undergo weight changes during fine-tuning, they remain consistently sensitive to safety.), enabling efficient safety transfers across models.
>
> 2. In contrast to [R3]'s parameter/activation steering and [R4]'s subspace fusion which require repeated computations per model, our approach uniquely **extracts a single Meta-SafetyLock from the base model that can be rapidly distributed** (0.0001s) to various fine-tuned variants without GPU resources or repeated safety pattern searches.
>
> 3. Our technical contribution is particularly evident in the **efficiency of distribution**. Where [R2] patches each model individually and [R4] requires subspace identification per model, Meta-SafetyLock's attention-head consistency property enables one-time extraction and rapid deployment across models without compromising safety or utility.

---

> ### Author Response · Authors · 2024-11-27
> **Response 2**
>
> In the discussion section, we have added comparisons with recent related work (NeurIPS 2024) [1], which uses similar representation engineering for experiments. We conducted further experiments with the base model (Llama-3-8B-Instruct) and across three different levels of fine-tuning scenarios. The results are as follows:
>
> **Level-1 Results:**
>
> | Model                              | AdvBench ASR | HEx-PHI Score |
> | ---------------------------------- | ------------ | ------------- |
> | Original Fine-tuned                | 49.24%       | 4.13          |
> | Circuit Breakers [NeurIPS 2024, 1] | 84.62%       | 3.62          |
> | SafetyLock                         | **0.19%**    | **1.36**      |
>
> **Level-2 Results:**
>
> | Model                              | AdvBench ASR | HEx-PHI Score |
> | ---------------------------------- | ------------ | ------------- |
> | Original Fine-tuned                | 38.46%       | 3.19          |
> | Circuit Breakers [NeurIPS 2024, 1] | 27.12%       | 2.10          |
> | SafetyLock                         | **5.19%**    | **1.07**      |
>
> **Level-3 Results:**
>
> | Model                              | AdvBench ASR | HEx-PHI Score |
> | ---------------------------------- | ------------ | ------------- |
> | Original Fine-tuned                | 42.88%       | 3.23          |
> | Circuit Breakers [NeurIPS 2024, 1] | 94.04%       | 3.79          |
> | SafetyLock                         | **0.19%**    | **1.04**      |
>
> Notably, NeurIPS 2024 method [1] requires more additional time (22 minutes 15 seconds on an A100). In contrast, SafetyLock requires only 5 minutes to construct Meta-SafetyLock and 0.1 seconds for distribution to any fine-tuned model. Furthermore, we evaluated both methods on the original paper's [4] benchmark scenarios using Llama-3-8B-Instruct:
>
> | Model                              | AutoDAN ASR | PAIR ASR | GCG ASR  |
> | ---------------------------------- | ----------- | -------- | -------- |
> | Base Model                         | 3.7%        | 18.7%    | 44.5%    |
> | Circuit Breakers [NeurIPS 2024, 1] | 0.0%        | 7.5%     | 2.5%     |
> | SafetyLock                         | **0.0%**    | **0.0%** | **0.0%** |
>
> Our framework represents a significant advancement in making safety alignment both theoretically grounded and practically scalable, while building upon the important foundational work of our predecessors. The effectiveness of this approach is demonstrated through our comprehensive evaluation across three distinct risk levels and novel dual-attack scenarios. We thank you again for raising this point. We have revised the Introduction section to highlight the distinctive features of our approach.
>
>
>
> ---
>
> [1] Improving Alignment and Robustness with Circuit Breakers

---

> ### Author Response · Authors · 2024-11-27
> **Response 3**
>
> >  The experiments demonstrating the utility of the method are limited. I understand that works like Qi et al. (2023b) [R0] largely rely on three risk categories, which is sufficient to prove that the problem of safety compromise due to fine-tuning exists. However, solving the problem would require more substantial evidence than relying on a similar set of experiments. It would be beneficial to see the method applied across different fine-tuning domains (such as math, code, and various other tasks) and observe the domain-specific performance with and without the safety measures. For instance, fine-tune on GSM8K, evaluate the test-set performance post task-specific tuning with and without the safety lock, and compare these results with those of related works.
>
> We sincerely appreciate your **thoughtful suggestion** to expand our experimental validation across different domains. Following your advice, we conducted additional experiments during the rebuttal period using Llama-3-8B-Instruct on the **GSM8K dataset**, evaluating both safety metrics and mathematical performance:
>
> | Model               | AdvBench ASR | HEx-PHI Score | GSM8K Test Acc |
> | ------------------- | ------------ | ------------- | -------------- |
> | Original            | 7.23%        | 1.45          | **85.59%**     |
> | Model-Edited (DINM) | 3.02%        | 1.33          | 5.00%          |
> | SafetyLock          | **0.19%**    | **1.08**      | 85.01%         |
>
> The results are particularly **encouraging**: SafetyLock maintains strong mathematical performance (only 0.68% drop in GSM8K accuracy) while **significantly enhancing safety** (97.4% reduction in AdvBench ASR). This demonstrates that SafetyLock effectively preserves domain-specific capabilities while ensuring robust safety guardrails. Based on your valuable feedback, we have added Appendix D.4 to provide comprehensive details about these domain-specific experiments, further strengthening our paper's empirical foundation.
>
>
>
> > Can you explain the need for the standard deviation of activations {\sigma}_{l}^h in equation 4? Any experiments that show with and without {\sigma}_{l}^h effect?
>
>
>
> We deeply appreciate your perceptive question about the necessity of including the standard deviation of activations, σlh{\sigma}_{l}^h, in Equation 4. This factor is fundamental in ensuring stable and effective interventions by normalizing the magnitude of adjustments to align with the variability of activations along the projection direction for each layer ll and head hh.
>
> To validate its importance, we conducted experiments comparing the performance of SafetyLock with and without σlh{\sigma}_{l}^h. The results are summarized below:
>
> | Model                         | AdvBench ASR | HEx-PHI Score | GSM8K Test Acc |
> | ----------------------------- | ------------ | ------------- | -------------- |
> | Original                      | 7.23%        | 1.45          | 85.59%         |
> | SafetyLock w/o {\sigma}_{l}^h | **0.0%**     | **1.03**      | 52.24%         |
> | SafetyLock w {\sigma}_{l}^h   | 0.19%        | 1.08          | **85.01%**     |
>
> For AdvBench ASR and HEx-PHI Score, lower values indicate better performance, while higher values are desirable for GSM8K Test Accuracy. Without σlh{\sigma}*{l}^h, while the ASR drops to 0.0% and the HEx-PHI Score improves slightly, there is a substantial degradation in accuracy on GSM8K (52.24%), undermining the model's utility. By incorporating σlh{\sigma}*{l}^h, the GSM8K accuracy recovers to 85.01%, close to the original model's performance, while maintaining significantly reduced ASR and an improved HEx-PHI Score compared to the baseline.
>
> These results highlight that σlh{\sigma}_{l}^h plays a vital role in balancing safety and accuracy, ensuring that interventions do not over-correct activations at the expense of model performance. Your thoughtful question encouraged us to reflect deeply on this design decision, and we are sincerely grateful for the opportunity to provide these insights. Thank you very much for your suggestion. We have added this experiment to Appendix D.5.

---

> ### Author Response · Authors · 2024-11-27
> **Response 4**
>
> > Can you please elaborate on section 4.5? The setting and motivation aren't very clear to me.
>
>
>
> We are deeply grateful for your astute observation about Section 4.5's clarity. Based on your valuable feedback, we have thoroughly revised this section to better articulate its **critical motivation** and experimental setup. The fundamental challenge in AI safety lies in what we call the **"alignment tax"** - the perceived trade-off between safety measures and model performance. For example, in the case of GSM8K mentioned above, training the model to refuse to answer all questions (by directly outputting "I cannot") can indeed reduce the ASR to 0 and result in the lowest HEx-PHI score of 1. However, this would also cause the test accuracy on the GSM8K dataset to drop to 0. While the simplest approach to ensuring safety would be making models reject all queries, this would render them useless. Instead, our **ideal objective** is to develop a system where models can effectively **reject harmful queries** while maintaining their **original performance levels** on legitimate tasks.
>
> SafetyLock achieves this delicate balance through its innovative approach, as demonstrated by our comprehensive evaluation across multiple benchmarks (AddSub, AQUA, CommonSenseQA, GSM8k, MT-Bench, and AlpacaEval 2.0). The results show that SafetyLock successfully maintains the model's performance (85.57% on AddSub compared to the original 86.33%) while significantly enhancing safety guardrails. This stands in stark contrast to existing safety methods, which often lead to **severe performance degradation** (e.g., Safe-Edited's complete performance collapse to 0.00%). These results underscore SafetyLock's unique contribution to the field: achieving robust safety without compromising the model's fundamental capabilities. Thank you for helping us improve the clarity of this crucial aspect of our work.

---

### Author Response · Authors · 2024-11-27
**General Responses and Summary of Revisions**

We sincerely thank all reviewers for their careful and constructive feedback, which has helped us significantly improve our work. We are encouraged by the reviewers' recognition of several key strengths:

- "The paper proposes a new idea to prevent safety compromise owing to downstream fine-tuning of the model" (*Review eTzc*)
- "Experiments show that SafetyLock significantly reduces harmful behavior" (*Review ZuM2*)
- "The paper provides insights into the inner mechanism of LLMs on safety" (*Review ZuM2*)
- "The experiments are sound, demonstrating the method's effectiveness in mitigating safety issues across various risk levels" (*Review eTzc*)

Based on your constructive comments, we have made substantial improvements to strengthen our paper's contributions and address the limitations. The major enhancements include:

1. **Comparison with Circuit Breakers (NeurIPS 2024)**: We have added comprehensive comparison with Circuit Breakers, including performance evaluation across three risk levels (Level-1: ASR from 84.62% to 0.19%, Level-2: 27.12% to 5.19%, Level-3: 94.04% to 0.19%), efficiency analysis (SafetyLock: 5min + 0.1s/model vs. Circuit Breakers: 22min/model), superior performance demonstration on their original benchmarks (AutoDAN, PAIR, GCG), SafetyLock achieved superior safety performance while maintaining 0% ASR across all scenarios. And detailed analysis in Appendix D.6.

1. **Citation and Related Work**: We have thoroughly expanded our literature review and comparisons with closely related work, added comprehensive comparisons with safety vector approaches (R1-R4 in Review eTzc), included recent works on safety mechanisms and neuron-level analysis, and removed claims of being "first" while properly positioning our contributions (Section 1, 2).
2. **Domain-Specific Experiments**: We have substantially expanded our experimental validation by adding GSM8K mathematical reasoning case study, including detailed performance analysis across different domains, and comparing domain-specific capabilities with and without SafetyLock (Appendix D.4).
3. **Activation Analysis**: We have enhanced our technical analysis by adding ablation studies on standard deviation term effects, including multi-head activation pattern analysis, and expanding intermediate layer analysis beyond the final layer (Section 3.2, Appendix D.3).
4. **Technical Clarifications**: We have added detailed analysis of learning rate impacts (1e-6 vs 2e-5), clarified token window size selection (r parameter), and expanded explanation of intervention degree (K) selection (Appendix D.1, D.2).
5. **Experimental Robustness**: We have added XSTest benchmark evaluation, included comprehensive baselines comparison, and enhanced analysis of safety direction consistency (Section 4.4, Table 2).
6. **Implementation Details**: We have added ablation studies on activation normalization, clarified the choice of preference-style safety dataset, and detailed the impact of token alignment in safety direction calculation (Appendix C, D).
7. **Open-Source Considerations**: We have added detailed discussion on deployment strategies, included hybrid approach combining transparency with controlled access, and expanded practical guidelines for implementation (Appendix E).
8. **Presentation Improvements**: We have fixed citation formatting and duplicates, improved figure captions and spacing, and enhanced overall clarity and readability throughout the paper.

We believe these comprehensive improvements address the key concerns raised while strengthening the paper's technical depth and practical impact. The additions substantially enhance the paper's contribution to the field of LLM safety alignment. We would be honored if the reviewers find these enhancements worthy of a more favorable evaluation. We welcome any additional feedback or questions from the reviewers.

---

### Meta-Review · Area_Chair_URTJ · 2024-12-19

**Metareview:**

The paper proposes SafetyLock, a method to maintain safety in fine-tuned LLMs by extracting safety bias directions from base models. While the work shows promising results in reducing harmful responses and introduces an efficient approach for safety restoration, several limitations exist. Key strengths include the discovery that fine-tuned models retain safety-related activation patterns and achieving sub-second safety restoration. Major weaknesses include limited applicability to open-source models, insufficient comparison with recent work like Circuit Breakers, and methodological concerns about token alignment in safety direction calculation. Given these significant limitations despite novel contributions, I recommend rejecting this paper.

**Additional Comments On Reviewer Discussion:**

During rebuttal, authors addressed reviewers' concerns about technical novelty by comparing with Circuit Breakers and demonstrating superior performance across risk levels. They provided ablation studies for parameter choices (r=5 tokens, std term) and safety head consistency. However, fundamental concerns about deployment for open-source models remain partially unresolved. The extensive experiments and analysis in rebuttal somewhat strengthen the paper but do not fully overcome its core limitations.

---

### Decision · Program_Chairs · 2025-01-22

Reject